

SciPost Phys. Lect. Notes 33 (2021)

# Cavity and replica methods for the spectral density of sparse symmetric random matrices

**Vito A R Susca, Pierpaolo Vivo⋆ and Reimer Kühn**

King's College London, Department of Mathematics,
Strand, London WC2R 2LS, United Kingdom

⋆ pierpaolo.vivo@kcl.ac.uk

## Abstract

We review the problem of how to compute the spectral density of sparse symmetric random matrices, i.e. weighted adjacency matrices of undirected graphs. Starting from the Edwards-Jones formula, we illustrate the milestones of this line of research, including the pioneering work of Bray and Rodgers using replicas. We focus first on the cavity method, showing that it quickly provides the correct recursion equations both for single instances and at the ensemble level. We also describe an alternative replica solution that proves to be equivalent to the cavity method. Both the cavity and the replica derivations allow us to obtain the spectral density via the solution of an integral equation for an auxiliary probability density function. We show that this equation can be solved using a stochastic population dynamics algorithm, and we provide its implementation. In this formalism, the spectral density is naturally written in terms of a superposition of local contributions from nodes of given degree, whose role is thoroughly elucidated. This paper does not contain original material, but rather gives a pedagogical overview of the topic. It is indeed addressed to students and researchers who consider entering the field. Both the theoretical tools and the numerical algorithms are discussed in detail, highlighting conceptual subtleties and practical aspects.

# 1  Introduction

The calculation of the average spectral density of eigenvalues of random matrices belonging to a certain ensemble has traditionally been one the fundamental problems in Random Matrix Theory (RMT), ever since the application of RMT to the statistics of energy levels of heavy nuclei [1]. The spectral problem has retained its centrality in RMT with diverse applications in physics [2], computer science [3], finance [4–6] and statistics [7,8]. The most celebrated results about the density of states such as the Wigner semicircle law [9] for Wigner matrices (including Gaussian ensembles) and the Marčenko-Pastur law [10] for covariance matrices refer to "dense" matrix ensembles, i.e. those for which most of the matrix entries are non-zero.

On the other hand, the spectral problem is very relevant also for "sparse" matrix models, i.e. when most of the entries are zero. Indeed, the spectral properties of (weighted) adjacency matrices of sparse graphs encode the structural and topological features of many complex

systems [11, 12]. For random walks and dynamical processes on graphs, the eigenvalue spectrum of the corresponding Markov transition matrix is directly connected to the relaxation time spectrum [13, 14]. Moreover, sparsely connected matrix models provide a test ground for physical systems described by Hamiltonians with finite-range interactions. In particular, tight-binding Hamiltonian operators with a kinetic term and an on-site random potential translate into matrix models that involve discrete graph Laplacians with additional random contributions to diagonals [15]. The spectra of such matrices have been used for the characterisation of many physical systems in condensed matter such as the study of gelation transitions in polymers [16]. Moreover, the behaviour of supercooled liquids can be described in terms of the spectrum a random sparse matrix representing the Hessian of those systems, within the framework of instantaneous normal modes [17].

Spectra of sparse random matrices and trees have also been employed as the simplest model to study *Anderson localisation* [18], i.e. the phenomenon by which a metal becomes an insulator due to disorder, such as impurities. The metallic phase corresponds to a spatially extended electronic wave function, allowing transport. On the other hand, a high level of disorder leads to localised wave functions, which prevent conduction. A model for this phase separation is represented by the *localisation transition* characterising the spectra of sparse random matrices, where eigenvalues related to delocalised eigenvectors are separated at the *mobility edge* from those related to localised eigenvectors (see Section 5). Localisation phenomena have been analysed on Bethe lattices [1] (see [15, 19] and the seminal paper of Abou-Chacra and collaborators [20], in which the cavity method that will be discussed below was also used) and on sparse random graphs [21–25].

In this paper, we describe the various strategies to compute the average spectral density (also known as density of states) for ensembles of sparse symmetric random matrices, i.e. weighted adjacency matrices of undirected graphs. Given a $N \times N$ random matrix $J$ with eigenvalues $\{\lambda_i\}_{i=1,\dots,N}$, the average spectral density is defined as

$$\rho(\lambda) = \left\langle \frac{1}{N} \sum_{i=1}^{N} \delta(\lambda - \lambda_i) \right\rangle_J , \qquad (1)$$

where the limit $N \to \infty$ is understood and $\langle \dots \rangle_J$ denotes the average over the matrix ensemble to which $J$ belongs. The latter is also referred to as "disorder" average. For a given (large) $N$, $\rho(\lambda)$ can be numerically obtained by first diagonalising a large number $M$ of $N \times N$ matrices drawn from the ensemble, collecting all their $N \cdot M$ eigenvalues and organising them into a normalised histogram. Our analysis is rooted in the statistical mechanics of disordered systems, with the main technical tools being the *cavity* (Section 3) and *replica* methods (Section 4 and 5).

## 1.1 A historical perspective on the spectral problem for sparse matrices

Our analysis will follow the historical developments that led to the solution of the problem. We start from the celebrated Edward-Jones formula [26], which is a key result linking the spectral problem to statistical mechanics. Indeed, the formula recasts the determination of the average spectral density (1) in terms of the average free energy $\langle \log Z(\lambda) \rangle_J$ of a disordered system with partition function $Z(\lambda)$. Edward and Jones were the first to use the *replica* method, extensively employed in spin-glass physics [27], to perform averages of this type in the context of random matrices.

Historically, the application of the Edwards-Jones recipe to sparse symmetric random matrices (in particular Erdős-Rényi adjacency matrices of graphs with finite mean degree $c$ and

---

[1]Bethe lattices are infinite regular trees.

the non-zero entries drawn from a Bernoulli distribution) was pioneered by Bray and Rodgers in [28] (and in a similar context in [29] and later on in [30]). However, in their formulation the evaluation of the average spectral density $\rho(\lambda)$ relies on the solution of a very complicated integral equation. The same integral equation has been derived independently with a supersymmetric approach in [31] and later obtained in a rigorous manner in [32], thus confirming the exactness of the symmetry assumptions in [28]. A full analytical solution of this equation is still unavailable. A numerical solution for large average connectivity $c$ was found in [17], whereas a solution in the form of an expansion for small $c$ was proposed in [16]. The difficulties in dealing with this equation stimulated the search for a variety of approximation schemes, such as large average connectivity expansions [28], the single defect approximation (SDA) [33] and the effective medium approximation (EMA) [34, 35]. Alongside approximation schemes, results from numerical diagonalisation such as in [36] have been employed to investigate the spectral properties of sparse random matrices.

A different approach to the spectral problem of sparse symmetric random matrices was proposed in [37]. There, the order parameters of the replica calculation are represented as uncountably infinite superpositions of Gaussians with random variances, as suggested by earlier solutions of models for finitely coordinated harmonically coupled systems [38]. A replica-symmetric Gaussian ansatz for the order parameter had appeared earlier in the random matrix context in [39], but was evaluated only within the SDA approximation. In [37], the intractable Bray-Rodgers integral equation is replaced by non-linear fixed-point equations for probability density functions, which are solved by a stochastic population dynamics algorithm. We will review both approaches in Sections 4 and 5 below.

Almost in parallel to [37], the *cavity* method [40] started to be employed for the determination of the spectral density of sparse symmetric random matrices by Rogers and collaborators in [41]. The cavity method, also known as *belief propagation*, represents a much simpler alternative to replicas and was originally introduced for the study of spectra of dense Lévy matrices in [42] and for diluted systems in [17]. The exactness of the cavity method for locally tree-like graphs with finite mean degree $c$ was proved in [43]. In [41], building on the Edwards-Jones setup, the authors used the cavity method to compute the spectrum of large single instances of sparse symmetric random matrices. The ensemble average spectral density (1) is then obtained building on the single-instance results, circumventing the calculation of the average "free energy" $\langle \log Z(\lambda) \rangle_J$ altogether. The cavity treatment produces non-linear fixed-point integral equations that are completely equivalent to those obtained in [37] within the replica framework.

It has been shown in [44] that both the cavity and the replica method yield the same results concerning the spectral density of graphs. Both approaches in [37] and [41] recover known results such as the Kesten-McKay law for the spectra of random regular graphs [45, 46], the Marčenko-Pastur law and Wigner's semicircle law respectively for sparse covariance matrices and for Erdős-Rényi adjacency matrices in the large mean degree limit. Moreover, both methods allow one to characterise the spectral density of sparse Markov matrices [47, 48] and graphs with modular [49] and small-world [50] structure and with topological constraints [51]. In a similar manner, both methods have also been employed to study the statistics of the top and second largest eigenpair of sparse symmetric random matrices [52–54]. The two methods have also been extended to the case of sparse non-Hermitian matrices [55–58]. A particular attention has been devoted to the spectral properties of the Hashimoto non-backtracking operator on random graphs [59, 60]. Both cavity and replica methods have been recently used to characterise the dense ($c \to \infty$) limit of the spectral density of adjacency matrices of undirected graphs within the configuration model, which reveals that the behaviour of the limiting spectral density is not universal but actually depends on degree fluctuations. Indeed the expected Wigner semicircle is recovered when the degree distribution tightly concentrates

around the mean degree $c$ for $c \to \infty$, whereas non trivial deviations from the semicircle are found when degree fluctuations are stronger [61].

Moreover, thanks to the extension of the replica method to the analysis of sparse loopy random graphs, the influence of loops on the spectra of sparse matrices has been lately investigated in [62, 63]. There is also a recent cavity analysis of the problem of loopy graphs by Newman and collaborators in [64].

## 1.2 Paper organisation

In this paper, we will retrace the main milestones in the determination of the spectral density of sparse symmetric random matrices. We start with the analysis of the Edwards-Jones formula in Section 2, providing its proof in Section 2.1 and discussing how to deal with the average $\langle \log Z(\lambda) \rangle_J$ in Section 2.2. For clarity and simplicity, we will first illustrate the cavity approach in Section 3. We outline the cavity setup in Section 3.1, then we deal with the spectrum of large instances of sparse symmetric random matrices in Section 3.2. In Section 3.3 we show how the single-instance approach can be extended to the $N \to \infty$ limit to recover the ensemble average spectral density. Besides, in 3.4 we evaluate the large $c$ limit of the average spectral density obtained within the cavity formalism, showing that it converges to the Wigner semicircle. We will then follow the historical development of the subject by documenting the Bray-Rodgers replica approach in Section 4. We will derive the Bray-Rodgers integral equation in Section 4.3, while in Section 4.4 we will obtain its large $c$ asymptotic expansion, showing that its leading order gives rise to the Wigner semicircle, as expected. In Section 5 we will deal with the alternative replica solution proposed in [37], showing in Section 5.1 that the solution obtained with this approach coincides with that found by the cavity treatment in Section 3.3. In Section 6, we outline the stochastic population dynamics algorithm employed to solve the non-linear fixed-point integral equations that are found within both the cavity and replica frameworks respectively in Section 3.3 and Section 5.1.

## 2 Edwards-Jones formula

Edwards and Jones in [26] provide a formula to express the average spectral density of $N \times N$ random matrices (1) as

$$\rho(\lambda) = -\frac{2}{\pi N} \lim_{\varepsilon \to 0^+} \mathrm{Im} \frac{\partial}{\partial \lambda} \langle \log Z(\lambda) \rangle_J \,, \tag{2}$$

with

$$Z(\lambda) = \int_{\mathbb{R}^N} \mathrm{d} \boldsymbol{v} \exp\left[ -\frac{\mathrm{i}}{2} \boldsymbol{v}^T (\lambda_\varepsilon \mathbb{1} - J) \boldsymbol{v} \right] \,, \tag{3}$$

where again the $\langle ... \rangle_J$ denotes the average over the matrix ensemble to which $J$ belongs. In (2), which is valid for any $N$, Im indicates the imaginary part and log is the branch of the complex logarithm for which $\log e^z = z$. In (3), the symbol $\mathbb{1}$ represents the $N \times N$ identity matrix, the symbol $\boldsymbol{v}$ describes a vector in $\mathbb{R}^N$ and the integral extends over $\mathbb{R}^N$. Moreover, $\lambda_\varepsilon = \lambda - \mathrm{i}\varepsilon$, where $\varepsilon$ is a positive parameter ensuring that the integral (3) is convergent, since the absolute value of the integrand has the leading behaviour $e^{-\frac{\varepsilon}{2} \sum_{i=1}^N v_i^2}$. The integral (3) can be interpreted as the canonical partition function of the Gibbs-Boltzmann distribution of $N$ harmonically coupled particles with an imaginary (inverse) temperature, viz.

$$P_J(\boldsymbol{v}) = \frac{1}{Z(\lambda)} \exp[-\mathrm{i} H(\boldsymbol{v})] \,, \tag{4}$$

with a complex "Hamiltonian"

$$H(\mathbf{v}) = \frac{1}{2}\mathbf{v}^T(\lambda_\epsilon \mathbb{1} - J)\mathbf{v} \, . \tag{5}$$

In this framework, the computation of (1) requires to evaluate $\langle \log Z(\lambda)\rangle_J$, which is the canonical free energy of the associated $N$ particles system, averaged over the random couplings.

## 2.1 Proof of the Edwards-Jones formula

The starting point is the definition (1). Looking for a representation of the Dirac delta, one considers the Sokhotski-Plemelj identity (see Appendix A for a proof), viz.

$$\frac{1}{x \pm i\varepsilon} \xrightarrow[\varepsilon\to0^+]{} \mathrm{Pr}\left(\frac{1}{x}\right) \mp i\pi\delta(x), \tag{6}$$

where $x \in \mathbb{R}$ and Pr denotes the Cauchy principal value. The imaginary part of the identity, namely

$$\delta(x) = \frac{1}{\pi}\lim_{\varepsilon\to0^+}\mathrm{Im}\frac{1}{x - i\varepsilon}\,, \tag{7}$$

provides the desired representation. Therefore, inserting (6) into (1) results in

$$\begin{aligned}
\rho(\lambda) &= \frac{1}{\pi N}\lim_{\varepsilon\to0^+}\mathrm{Im}\left\langle\sum_{i=1}^{N}\frac{1}{\lambda - \lambda_i - i\varepsilon}\right\rangle_J \\
&= -\frac{1}{\pi N}\lim_{\varepsilon\to0^+}\mathrm{Im}\left\langle\sum_{i=1}^{N}\frac{1}{\lambda_i + i\varepsilon - \lambda}\right\rangle_J,
\end{aligned} \tag{8}$$

where the minus sign has been made explicit.

One would now express the ratio in the angle brackets as the derivative of the *principal branch* of the complex logarithm, denoted by Log. Unlike other properties, its derivative behaves exactly like that of the real logarithm, therefore

$$\sum_{i=1}^{N}\frac{1}{\lambda_i + i\varepsilon - \lambda} = -\frac{\partial}{\partial\lambda}\sum_{i=1}^{N}\mathrm{Log}\,(\lambda_i + i\varepsilon - \lambda)\,, \tag{9}$$

entailing for the average spectral density the formula

$$\rho(\lambda) = \frac{1}{\pi N}\lim_{\varepsilon\to0^+}\mathrm{Im}\frac{\partial}{\partial\lambda}\left\langle\sum_{i=1}^{N}\mathrm{Log}\,(\lambda_i + i\varepsilon - \lambda)\right\rangle_J. \tag{10}$$

The sum of logarithms in (10) can be related to the partition function $Z(\lambda)$ in (3) by exploiting the following identity [65, 66],

$$Z(\lambda) = \int_{\mathbb{R}^N}d\mathbf{v}\,\exp\left[-\frac{i}{2}\mathbf{v}^T(\lambda_\varepsilon\mathbb{1} - J)\mathbf{v}\right] = (2\pi)^{N/2}\exp\left[-\frac{1}{2}\sum_{i=1}^{N}\mathrm{Log}\,(\lambda_i + i\varepsilon - \lambda) + \frac{i\pi N}{4}\right]. \tag{11}$$

Caution is needed when taking the logarithm on both sides of (11), as in general $\mathrm{Log}(e^z) \neq z$ (see Appendix B). Indeed, using the property (172) and taking the principal logarithm on both sides of (11), one would obtain

$$\sum_{i=1}^{N}\mathrm{Log}\,(\lambda_i + i\varepsilon - \lambda) = -2\mathrm{Log}Z(\lambda) + N\mathrm{Log}(2\pi) + \frac{i\pi N}{2} + 4\pi i\left\lfloor\frac{1}{2} - \frac{g(\lambda)}{2\pi}\right\rfloor, \tag{12}$$

where

$$g(\lambda) = -\frac{1}{2}\sum_{i=1}^{N}\mathrm{Arg}(\lambda_i + \mathrm{i}\varepsilon - \lambda) + \frac{\pi N}{4} \tag{13}$$

is the imaginary part of the exponent in (11) and the symbol $\lfloor ... \rfloor$ denotes the floor operation, i.e. $\lfloor x \rfloor$ is the integer such that $x - 1 < \lfloor x \rfloor \leq x$ for $x \in \mathbb{R}$.

Note that this branch choice would make the r.h.s. not everywhere differentiable for $\lambda \in \mathbb{R}$. Therefore, it is convenient to pick the branch of the complex logarithm such that $\log e^z = z$ instead, i.e. for which the extra (non-differentiable) phase term in (12) is killed. This choice yields

$$\sum_{i=1}^{N}\mathrm{Log}(\lambda_i + \mathrm{i}\varepsilon - \lambda) = -2\log Z(\lambda) + N\log(2\pi) + \frac{\mathrm{i}\pi N}{2}, \tag{14}$$

where the constant terms on the r.h.s. depend on $N$, but not on $\lambda$. Taking the derivative, one eventually finds

$$\frac{\partial}{\partial\lambda}\sum_{i=1}^{N}\mathrm{Log}(\lambda_i + \mathrm{i}\varepsilon - \lambda) = -2\frac{\partial}{\partial\lambda}\log Z(\lambda), \tag{15}$$

therefore the Edwards-Jones formula (2) is recovered.

## 2.2 Tackling the average in the Edwards-Jones formula

In order to obtain the spectral density, the average $\langle \log Z(\lambda)\rangle_J$ must be computed. It explicitly reads

$$\langle \log Z(\lambda)\rangle_J = \int\prod_{i<j}\mathrm{d}J_{ij}P(\{J_{ij}\})\log\int_{\mathbb{R}^N}\mathrm{d}\boldsymbol{v}\,\exp\left[-\frac{\mathrm{i}}{2}\boldsymbol{v}^T(\lambda_\varepsilon\mathbb{1} - J)\boldsymbol{v}\right], \tag{16}$$

where $P(\{J_{ij}\})$ is the joint distribution of the matrix entries. The presence of the logarithm in (16) prevents a factorisation of averages over edges $(i,j)$ even for a factorised pdf of the $J_{ij}$. The only available strategy seems to perform the inner $N$-fold integral over $\boldsymbol{v}$ first, compute the logarithm, and then average over the random matrix disorder. However, this sequence of operations would simply run the Edwards-Jones formula (2) backwards, leading to the useless identity $\rho(\lambda) = \rho(\lambda)$. The only chance to make some progress therefore relies on performing the disorder average *first*. However, the two integrations in (16) cannot be directly exchanged due to the presence of the logarithm in between.

Disorder averages such as (16) are called *quenched* averages. The technique to handle such averages is the *replica trick*. It is a well established method employed in the statistical mechanics of disordered systems that allows one to bypass the logarithm in (16) in favour of the computation of integer moments of $Z(\lambda)$ (see Section 4)[1].

The replica method for the calculation of the spectral density of dense random matrices was employed by Edwards and Jones in [26]. The same replica calculation for sparse ensembles was pioneered by Bray and Rodgers in [28]. However, we prefer to start with the cavity approach because it is technically much less involved and allows one to circumvent the direct computation of $\langle \log Z(\lambda)\rangle_J$. We will then follow the historical path traced in [28] in Section 4.

---

[1]There exists also an alternative though only approximate strategy, known as *annealed* average, which does not rely on the replica method. It consists in "moving" the logarithm outside the disorder average. Although formally incorrect, the annealed protocol provides the correct spectral density of "dense" random matrices, such as Gaussian ones (see Section 15.4 in [66] for a thorough discussion).

# 3 Cavity method for the spectral density

The cavity method as implemented in [41] makes it possible to derive the spectral density for a single instance of a large sparse symmetric matrix. According to the physical interpretation of the Edwards-Jones formula, the calculation of the spectral density can be recast as a problem of interacting particles on a sparse graph. The basic idea behind the cavity method [40] is that observables related to a certain node of a network in which cycles are scarce (thereby called *tree-like*) can be determined from the same network where the node in question is removed. Due to the sparse structure, the removal of a node makes its neighbouring sites (as well as the signals coming from them) uncorrelated.

## 3.1 Definition of the sparse matrix ensemble

We consider a large $N \times N$ sparse symmetric random matrix $J$. It represents the weighted adjacency matrix of an undirected graph $\mathcal{G}$, i.e. each entry can be expressed as $J_{ij} = c_{ij}K_{ij}$, where the $c_{ij} = c_{ji} \in \{0,1\}$ represent the pure adjacency matrix and the $K_{ij}$ encode the bond weights. When two nodes $i$ and $j$ are connected by a link, then $c_{ij} = 1$, otherwise $c_{ij} = 0$. We consider simple graphs, in which self-loops are not present, entailing that $c_{ii} = 0$ for any node $i$. In an undirected graph, the degree $k_i$ of the node $i$ is defined as the number of nodes in its neighbourhood $\partial i = \{j : c_{ij} = 1\}$, viz.

$$k_i = \sum_{j \in \partial i} c_{ij} = |\partial i| . \tag{17}$$

We define $c = \frac{1}{N} \sum_{i=1}^{N} k_i$ as the mean degree. We consider locally tree-like sparse matrices, in which the probability of finding a cycle vanishes as $\ln N/N$ when $N \to \infty$. Alternatively, this property is implied by the requirement that the mean degree $c$ does not increase with the matrix size $N$, hence $c/N \to 0$ as $N \to \infty$. In this very sparse regime, the cavity method predictions are approximate for sparse graphs of finite size $N$, whereas they are exact for finite trees. However, the cavity results become asymptotically exact on finitely connected networks in the limit $N \to \infty$ (i.e. in the thermodynamic limit). This has been rigorously proved in [43].

Following the statistical mechanics analogy, in the sparse case the $N$ particles described by the variables $v_i$ interact on the graph $\mathcal{G}$ where an edge is defined for any pair $(i, j)$ of interacting particles. While the replica formalism analyses the partition function (3) in the limit $N \to \infty$, the cavity method focusses on the associated Gibbs-Boltzmann distribution (4) with imaginary inverse temperature i and complex Hamiltonian (5), as shown in the section below.

## 3.2 Cavity derivation for single instances

The spectral density of $J$ is obtained from the Edwards-Jones formula (2) for finite $N$ as

$$\rho_J(\lambda) = -\frac{2}{\pi N} \lim_{\varepsilon \to 0^+} \text{Im} \frac{\partial}{\partial \lambda} \log Z(\lambda), \tag{18}$$

where $Z(\lambda)$ is defined in (3). The subscript indicates that $\rho_J(\lambda)$ refers to a single, specific instance $J$. For the same reason, no averaging is needed. Performing explicitly the $\lambda$-derivative in (18) with $Z(\lambda)$ defined in (3), one obtains

$$\frac{\partial}{\partial \lambda} \log Z(\lambda) = -\frac{i}{2} \sum_{i=1}^{N} \int \prod_{j=1}^{N} dv_j \, P_J(\boldsymbol{v}) v_i^2 , \tag{19}$$

where

$$P_J(\boldsymbol{v}) = \frac{1}{Z(\lambda)} \exp\left[ -\frac{i}{2} \boldsymbol{v}^T (\lambda_\epsilon \mathbb{1} - J) \boldsymbol{v} \right] \tag{20}$$

is the Gibbs-Boltzmann distribution defined in (4). For any given $i$, the average w.r.t. the joint pdf (20) in (19) reduces to the average w.r.t. the single-site marginal $P_i(v_i)$, viz.

$$\int \prod_{j=1}^{N} dv_j \, P_J(\boldsymbol{v}) v_i^2 = \int dv_i P_i(v_i) v_i^2 = \langle v_i^2 \rangle , \qquad (21)$$

where the $\langle v_i^2 \rangle$ represent the single-site variances of each of the $N$ marginal pdfs $P_i(v_i)$. Using (19) and (21), the spectral density in (18) can thus be written as

$$\rho(\lambda) = -\frac{2}{\pi N} \lim_{\varepsilon \to 0^+} \text{Im}\left( -\frac{i}{2} \sum_{i=1}^{N} \langle v_i^2 \rangle \right) = \frac{1}{\pi N} \lim_{\varepsilon \to 0^+} \sum_{i=1}^{N} \text{Re} \langle v_i^2 \rangle . \qquad (22)$$

Therefore, it is sufficient to determine the $N$ single-site variances to calculate the spectral density using (22).

In order to find the $\langle v_i^2 \rangle$, one looks at each marginal pdf $P_i(v_i)$. Due to the sparse nature of $J$, the variable $v_i$ is coupled (through $J_{ij}$) only to those $v_j$ associated to nodes that are neighbours of $i$. Hence, the single-site marginal of the node $i$ can be expressed as

$$P_i(v_i) = \int \prod_{j(\neq i)}^{N} dv_j \, P_J(\boldsymbol{v}) = \frac{1}{Z_i} e^{-\frac{i}{2}\lambda_\varepsilon v_i^2} \int d\boldsymbol{v}_{\partial i} e^{i \sum_{j \in \partial i} J_{ij} v_i v_j} P^{(i)}(\boldsymbol{v}_{\partial i}) . \qquad (23)$$

In (23), the integration is over the "particles" interacting with particle $i$, i.e. those sitting on the neighbouring sites $\partial i$. The distribution $P^{(i)}(\boldsymbol{v}_{\partial i})$ collects the contributions coming from the interaction of each of the $v_j$ ($j \in \partial i$) with particles sitting on nodes that are not neighbours of $i$ themselves (see Graph 1 on the l.h.s. of Fig. 1). The contributions to the integral defining $P_i(v_i)$ coming from nodes further away generate a constant term that is absorbed in the normalisation constant $Z_i$.

The distribution $P^{(i)}(\boldsymbol{v}_{\partial i})$ is called the *cavity* distribution, since it refers to a graph in which the node $i$ has been removed. In a tree-like structure, the neighbouring sites of each node $i$ are correlated mainly through the node $i$. Hence, when the node $i$ is removed, its neighbours become uncorrelated (see Graph 2 on the r.h.s. of Fig. 1). Therefore, the joint cavity pdf $P^{(i)}(\boldsymbol{v}_{\partial i})$ factorises into the product of independent *cavity marginals* $P_j^{(i)}(v_j)$, i.e.

$$P^{(i)}(\boldsymbol{v}_{\partial i}) = \prod_{j \in \partial i} P_j^{(i)}(v_j) . \qquad (24)$$

We remark that the condition (24) is exact only as $N \to \infty$, while being only approximate for finite $N$. From Eq. (24), it follows that the single-site marginal (23) can be expressed as

$$P_i(v_i) = \frac{1}{Z_i} e^{-\frac{i}{2}\lambda_\varepsilon v_i^2} \prod_{j \in \partial i} \int dv_j e^{i J_{ij} v_i v_j} P_j^{(i)}(v_j) . \qquad (25)$$

Eq. (25) shows that the marginal $P_i(v_i)$ is defined in terms of the cavity marginals $P_j^{(i)}(v_j)$. A self-consistent definition of each of the cavity marginal distributions $P_j^{(i)}(v_j)$ can be obtained by iterating the same reasoning as above. Indeed, one can now choose one of the nodes $j \in \partial i$ and define the marginal pdf associated to that node in the same way as in Eq. (25). However, the network one is considering at this stage is that where the node $i$ has already been removed, therefore eventually obtaining the *cavity marginal* $P_j^{(i)}(v_j)$, namely

$$P_j^{(i)}(v_j) = \frac{1}{Z_j^{(i)}} e^{-\frac{i}{2}\lambda_\varepsilon v_j^2} \prod_{\ell \in \partial j \backslash i} \int dv_\ell e^{i J_{j\ell} v_j v_\ell} P_\ell^{(j)}(v_\ell) , \qquad (26)$$

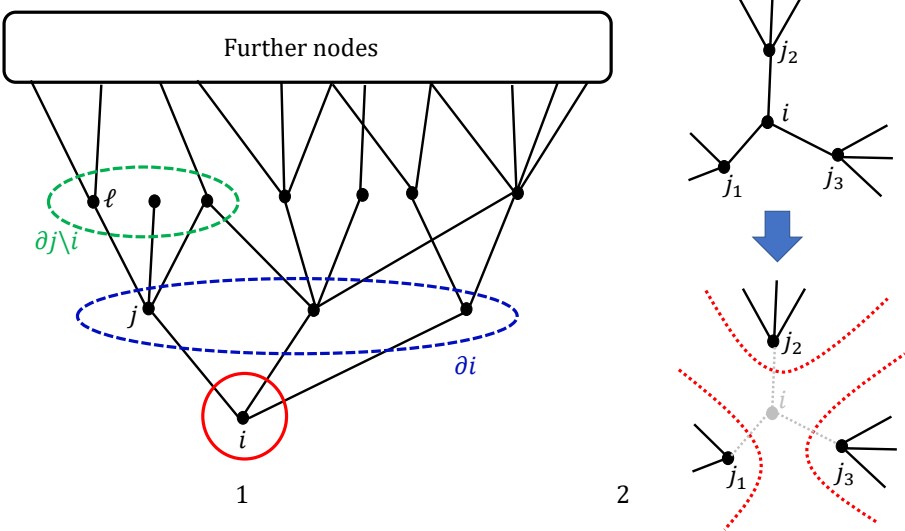

Figure 1: Graph sketches. **Graph 1**: a tree-like graph where the indices refer to the notation used in Section 3.1 to derive cavity single-instance equations. **Graph 2**: example of the decorrelation occurring to the nodes $j_1$, $j_2$ and $j_3$, neighbours of the node $i$, after the removal of $i$.

where the symbol $\partial j \backslash i$ denotes the set of neighbours of node $j$ excluding $i$ (see again Graph 1 on the l.h.s. of Fig. 1 ). In turn, the cavity marginals $P_\ell^{(j)}(v_\ell)$ are defined on the graph where also the node $j \in \partial i$ has been removed.

Eq. (26) defines a set of recursion equations for any pair of interacting nodes $(i, j)$. The set of recursion equations (26) is solved exactly by a zero-mean Gaussian ansatz for the cavity marginals $P_j^{(i)}(v_j)$. Indeed, assuming that

$$P_j^{(i)}(v_j) = \sqrt{\frac{\omega_j^{(i)}}{2\pi}} \exp\left(-\frac{\omega_j^{(i)}}{2} v_j^2\right), \tag{27}$$

and performing the Gaussian integrals on the r.h.s. of (26), one gets

$$P_j^{(i)}(v_j) = \frac{1}{Z_j^{(i)}} \exp\left[-\frac{1}{2}\left(i\lambda_\varepsilon + \sum_{\ell \in \partial j \backslash i} \frac{J_{j\ell}^2}{\omega_\ell^{(j)}}\right) v_j^2\right]. \tag{28}$$

The comparison between the exponents of (27) and (28) entails

$$\omega_j^{(i)} = i\lambda_\varepsilon + \sum_{\ell \in \partial j \backslash i} \frac{J_{j\ell}^2}{\omega_\ell^{(j)}}. \tag{29}$$

Therefore, the set of equations (26) translates into a set of self-consistency equations for the cavity inverse variances $\omega_j^{(i)}$.

Similarly, the Gaussian ansatz (27) can be inserted in the single-site marginal expression (25), yielding a Gaussian structure for $P_i(v_i)$, viz.

$$P_i(v_i) = \frac{1}{Z_i} \exp\left(-\frac{1}{2}\omega_i v_i^2\right), \tag{30}$$

with single-site inverse variances given by

$$\omega_i = i\lambda_\varepsilon + \sum_{j \in \partial i} \frac{J_{ij}^2}{\omega_j^{(i)}} \, . \tag{31}$$

Once the cavity inverse variances are determined as the solution of (29), the single-site inverse variances $\langle v_i^2 \rangle = \frac{1}{\omega_i}$ are found from (31), and the spectral density is readily obtained from (22) as

$$\rho_J(\lambda) = \frac{1}{\pi N} \lim_{\varepsilon \to 0^+} \sum_{i=1}^{N} \mathrm{Re}\left[\frac{1}{\omega_i}\right] = \frac{1}{\pi N} \lim_{\varepsilon \to 0^+} \sum_{i=1}^{N} \frac{\mathrm{Re}[\omega_i]}{(\mathrm{Re}[\omega_i])^2 + (\mathrm{Im}[\omega_i])^2} \, . \tag{32}$$

The formula (32) is exact in the limit $N \to \infty$: however for a finite but sufficiently large $N$, it provides an approximation for the average spectral density of the ensemble. As a concluding remark, it can be noticed that the set of self-consistency equations for the cavity inverse variances (29) only depends on the square of matrix entries, thus entailing that the spectrum of the matrix $J$ is equal to that of the matrix $-J$ and therefore is perfectly symmetric around $\lambda = 0$. This property indeed holds exactly for trees, since every tree is a bipartite graph (see [67] or Appendix G for a simple proof of this property). This check further corroborates that cavity equations are exact on trees, but only approximate on tree-like structures as long as cycles are negligible.

The set of cavity recursions (29) can be solved by a forward iteration algorithm. A working example of a code that allows one to determine the average spectral density on a single instance is available upon request.

## 3.3 Thermodynamic limit within the cavity framework

In this section we depart from [41] and show that the ensemble average of the spectral density (1) can be recovered from the single-instance spectral density (32) as obtained through the cavity method. Indeed, by invoking the law of large number in (32), in the large $N$ limit one gets

$$\rho_J(\lambda) = \frac{1}{\pi N} \lim_{\varepsilon \to 0^+} \sum_{i=1}^{N} \mathrm{Re}\left[\frac{1}{\omega_i}\right] \xrightarrow[N \to \infty]{} \rho(\lambda) = \frac{1}{\pi} \lim_{\varepsilon \to 0^+} \int d\tilde{\omega}\, \tilde{\pi}(\tilde{\omega}) \mathrm{Re}\left[\frac{1}{\tilde{\omega}}\right], \tag{33}$$

where $\tilde{\pi}(\tilde{\omega})$ is the pdf of the inverse variances $\omega_i$ taking values around $\tilde{\omega}$. In the r.h.s. of Eq. (33) the subscript $J$ has been dropped, as the quantity $\rho(\lambda)$ characterises the ensemble of $J$, rather than a single matrix. Eq. (33) implicitly assumes that the spectral density enjoys the *self-averaging* property, meaning that a large single instance of the ensemble faithfully represents the average behaviour over many instances.

The task now is to find the pdf of the inverse variances $\tilde{\pi}(\tilde{\omega})$. Recalling the single-instance relation (31) between the single-site inverse variances $\omega_i$ and the cavity inverse variances $\omega_j^{(i)}$, the pdf $\tilde{\pi}(\tilde{\omega})$ will be determined in terms of the probability density $\pi(\omega)$ of $\omega_j^{(i)}$.

In order to find the pdf $\pi(\omega)$, one observes that the set of self-consistency equations for the cavity inverse variances (29) refers to the *links* of the underlying graph. In an infinitely large network, links can be distinguished from one another by the degree of the node they are pointing to. Therefore, considering a link $(i, j)$ pointing to a node $j$ of degree $k$, the value $\omega$ of the cavity inverse variance $\omega_j^{(i)}$ living on this link is determined by the set $\{\omega_\ell\}_{k-1}$ of the $k-1$ values of the cavity inverse variances $\omega_\ell^{(j)}$ living on each of the edges connecting $j$ with its neighbours $\ell \in \partial j \backslash i$. In an infinite system, these values can be regarded as $k-1$ independent realisations of the random variables of type $\omega_\ell^{(j)}$, each drawn from the same pdf $\pi(\omega)$. The entries of $J$ appearing in (29) are replaced by a set $\{K_\ell\}_{k-1}$ of $k-1$ independent realisations

of the random variables $K_{j\ell}$, each distributed according to the bond weights pdf $p_K(K)$. The distribution $\pi(\omega)$ is then obtained by averaging the contributions coming from every link w.r.t. the probability $\frac{k}{c}p(k)$ of having a link pointing to a node of degree $k$[2]. This reasoning leads to the self-consistency equation

$$\pi(\omega) = \sum_{k=1}^{\infty} p(k)\frac{k}{c} \int \{\mathrm{d}\pi\}_{k-1} \left\langle \delta\left(\omega - \left(\mathrm{i}\lambda_\varepsilon + \sum_{\ell=1}^{k-1}\frac{K_\ell^2}{\omega_\ell}\right)\right)\right\rangle_{\{K\}_{k-1}}, \tag{34}$$

where $\{\mathrm{d}\pi\}_{k-1} = \prod_{\ell=1}^{k-1}\mathrm{d}\omega_\ell\pi(\omega_\ell)$ and the angle brackets $\langle\cdot\rangle_{\{K\}_{k-1}}$ denote the average over $k-1$ independent realisations of the random variable $K$. Eq. (34) is generally solved via a population dynamics algorithm (see Section 6).

The same reasoning can be applied to find the pdf $\tilde\pi(\tilde\omega)$ of inverse variances. Recalling (31), it can be noticed that the $\omega_i$ are variables related to *nodes*, rather than links. Since in the infinite size limit the nodes can be distinguished from one another by their degree, the pdf $\tilde\pi(\tilde\omega)$ can be written in terms of (34) as

$$\tilde\pi(\tilde\omega) = \sum_{k=0}^{\infty} p(k) \int \{\mathrm{d}\pi\}_k \left\langle \delta\left(\tilde\omega - \left(\mathrm{i}\lambda_\varepsilon + \sum_{\ell=1}^{k}\frac{K_\ell^2}{\omega_\ell}\right)\right)\right\rangle_{\{K\}_k}, \tag{35}$$

where $p(k)$ is the degree distribution.

Inserting (35) into (33) gives (at the ensemble level)

$$\rho(\lambda) = \frac{1}{\pi}\lim_{\varepsilon\to 0}\sum_{k=0}^{\infty} p(k)\mathrm{Re}\int \{\mathrm{d}\pi\}_k \left\langle \frac{1}{\mathrm{i}\lambda_\varepsilon + \sum_{\ell=1}^{k}\frac{K_\ell^2}{\omega_\ell}}\right\rangle_{\{K\}_k}$$

$$= \frac{1}{\pi}\lim_{\varepsilon\to 0^+}\sum_{k=0}^{\infty} p(k)\int \{\mathrm{d}\pi\}_k \left\langle \frac{\mathrm{Re}\left[\sum_{\ell=1}^{k}\frac{K_\ell^2}{\omega_\ell}\right] + \varepsilon}{\left(\mathrm{Re}\left[\sum_{\ell=1}^{k}\frac{K_\ell^2}{\omega_\ell}\right] + \varepsilon\right)^2 + \left(\lambda + \mathrm{Im}\left[\sum_{\ell=1}^{k}\frac{K_\ell^2}{\omega_\ell}\right]\right)^2}\right\rangle_{\{K\}_k}. \tag{36}$$

Eq. (36) is the ensemble generalisation of the single-instance formula (32) and provides the ensemble average of the spectral density (1). The average spectral density as expressed in (36) can be interpreted as a weighted sum of local densities, each pertaining to sites of degree $k$. As shown by (36), the solution of the spectral problem is *completely determined* by the distribution $\pi$ satisfying the self-consistency equation (34). Once $\pi$ has been obtained, the average spectral density (36) is evaluated by *sampling* from a large population representing the distribution $\pi(\omega)$. Section 6 illustrates the algorithm that produces the solution of self-consistency equations of this type as well as the details of the sampling procedure.

## 3.4 The $c \to \infty$ limit in the cavity formalism

One can easily show that taking the $c \to \infty$ limit in Eq. (34), (35) and then eventually (33), the Wigner semicircle law is recovered. This has been first shown in [41]. According to [61], we consider graphs in the configuration model having a degree distribution such that $\frac{\sigma_k^2}{\langle k\rangle^2} = \frac{\langle k^2\rangle - \langle k\rangle^2}{\langle k\rangle^2} \to 0$ as $\langle k\rangle = c \to \infty$. Here, the symbol $\sigma_k$ denotes the standard deviation of the degree distribution $p(k)$. For example, $\sigma_k = \sqrt{c}$ for Erdős-Rényi graphs.

---

[2]It can be observed that in general the probability that a node of degree $k$ is connected to a node of degree $k'$ is conditional, namely $P(k'|k)$. However, configuration model ensembles (including the Erdős-Rényi ensemble) are cases of random uncorrelated networks, hence $P(k'|k)$ is independent of $k$. Therefore, $P(k'|k)$ reduces to the probability that an edge points to a node of degree $k'$, which can be defined as the ratio between the number of edges pointing to nodes of degree $k'$, $k'p(k')$, and the number of edges pointing to nodes of any degree, i.e. the sum $\sum_{k'}k'p(k') = c$.

A meaningful large-$c$ limit is obtained for Eq. (34) or equivalently (35) by rescaling each instance of the bond random weights as $K_{ij} = \mathcal{K}_{ij}/\sqrt{c}$. Therefore, considering (34) one obtains

$$\pi(\omega) = \sum_{k=1}^{\infty} p(k) \frac{k}{c} \int \{d\pi\}_{k-1} \left\langle \delta\left(\omega - \left(i\lambda_\varepsilon + \frac{1}{c}\sum_{\ell=1}^{k-1} \frac{\mathcal{K}_\ell^2}{\omega_\ell}\right)\right)\right\rangle_{\{\mathcal{K}\}_{k-1}}. \tag{37}$$

For large $c$, the sum over the degrees in Eq. (37) receives contributions only from $k = c \pm \mathcal{O}(\sigma_k)$. As $c \to \infty$, the degree distribution $p(k)$ becomes highly concentrated around $k = c$, thus the argument of the $\delta$-function on the r.h.s of Eq. (37) can be evaluated using the Law of Large Numbers (LLN). Indeed, one finds that the r.h.s. of the condition

$$\omega = i\lambda_\varepsilon + \frac{1}{c}\sum_{\ell=1}^{c-1} \frac{\mathcal{K}_\ell^2}{\omega_\ell} \tag{38}$$

does not fluctuate, hence $\omega$ itself is fixed and determined by the algebraic equation

$$\bar{\omega}_\varepsilon = i\lambda_\varepsilon + \frac{\langle \mathcal{K}^2 \rangle}{\bar{\omega}_\varepsilon} \Leftrightarrow \bar{\omega}_\varepsilon = \frac{i\lambda_\varepsilon \pm \sqrt{4\langle \mathcal{K}^2 \rangle - \lambda_\varepsilon^2}}{2}. \tag{39}$$

For large $c$, the quantity $\langle \mathcal{K}^2 \rangle = \frac{1}{c}\sum_{\ell=1}^{c-1} \mathcal{K}_\ell^2 \simeq \frac{1}{c}\sum_{\ell=1}^{c} \mathcal{K}_\ell^2$ represents the second moment of the pdf of the rescaled bond weights.

The very same reasoning can be applied to the argument of the $\delta$ function in (35), entailing that in the limit $c \to \infty$ the $\tilde{\omega}$ are non-fluctuating as well, and take the same constant values given by the solutions of Eq. (39), viz.

$$\tilde{\pi}(\tilde{\omega}) = \delta(\tilde{\omega} - \bar{\omega}_\varepsilon) \quad \text{as } c \to \infty. \tag{40}$$

Therefore, inserting Eq. (39) and (40) in Eq. (33), one finds that in the limit $c \to \infty$

$$\begin{aligned}
\rho(\lambda) &= \frac{1}{\pi} \lim_{\varepsilon \to 0^+} \mathrm{Re}\left[\frac{1}{\bar{\omega}_\varepsilon}\right] \\
&= \frac{1}{\pi} \lim_{\varepsilon \to 0^+} \mathrm{Re}\left[\frac{2}{i\lambda_\varepsilon \pm \sqrt{4\langle \mathcal{K}^2 \rangle - \lambda_\varepsilon^2}}\right] \\
&= \frac{1}{2\pi\langle \mathcal{K}^2 \rangle} \lim_{\varepsilon \to 0^+} \mathrm{Re}\left[i\lambda_\varepsilon \mp \sqrt{4\langle \mathcal{K}^2 \rangle - \lambda_\varepsilon^2}\right],
\end{aligned} \tag{41}$$

which in the $\varepsilon \to 0^+$ limit eventually reduces to

$$\rho(\lambda) = \begin{cases} \frac{1}{2\pi\langle \mathcal{K}^2 \rangle}\sqrt{4\langle \mathcal{K}^2 \rangle - \lambda^2} & -2\sqrt{\langle \mathcal{K}^2 \rangle} < \lambda < 2\sqrt{\langle \mathcal{K}^2 \rangle} \\ 0 & \text{elsewhere} \end{cases}, \tag{42}$$

where the plus sign has been chosen to get a physical solution. The latter expression corresponds to the Wigner's semicircle.

## 3.5 The spectral density and the resolvent

Before dealing with the replica derivation of the average spectral density, it is worth remarking that the average spectral density can be obtained in an alternative way considering the *resolvent*. Given a $N \times N$ matrix $J$, its resolvent is defined as

$$G(z) = (z\mathbb{1} - J)^{-1}, \tag{43}$$

where $z \in \mathbb{C}$ and the matrix $\mathbb{1}$ is the $N \times N$ identity matrix. Setting $z = \lambda_\varepsilon = \lambda - i\varepsilon$, the average spectral density is obtained from the imaginary part of the trace of the resolvent matrix, i.e.

$$\rho(\lambda) = \lim_{\varepsilon \to 0^+} \frac{1}{\pi N} \operatorname{Im} \operatorname{Tr} \left\langle (\lambda_\varepsilon \mathbb{1} - J)^{-1} \right\rangle_J, \tag{44}$$

where the thermodynamic limit $N \to \infty$ is understood.

Eq. (44) can be explained by observing that the resolvent provides a regularised version of the Dirac delta appearing in Eq. (1). Indeed, the matrix $G$ shares the same eigenvector basis with $J$, $\{u_\alpha\}$ with $\alpha = 1, \ldots, N$. Then, using the spectral theorem, the resolvent (43) can be written as

$$G(\lambda - i\varepsilon) = \sum_{\alpha=1}^{N} \frac{1}{\lambda - i\varepsilon - \lambda_\alpha} u_\alpha u_\alpha^T, \tag{45}$$

entailing that

$$
\begin{aligned}
\rho(\lambda) &= \lim_{\varepsilon \to 0^+} \frac{1}{\pi N} \left\langle \operatorname{Im} \sum_{i=1}^{N} G(\lambda - i\varepsilon)_{ii} \right\rangle_J \\
&= \lim_{\varepsilon \to 0^+} \frac{1}{\pi N} \left\langle \operatorname{Im} \sum_{i=1}^{N} \sum_{\alpha=1}^{N} \frac{1}{\lambda - i\varepsilon - \lambda_\alpha} u_{i\alpha}^2 \right\rangle_J \\
&= \lim_{\varepsilon \to 0^+} \frac{1}{\pi N} \left\langle \sum_{\alpha=1}^{N} \operatorname{Im} \frac{1}{\lambda - i\varepsilon - \lambda_\alpha} \right\rangle_J \\
&= \frac{1}{N} \left\langle \sum_{\alpha=1}^{N} \delta(\lambda - \lambda_\alpha) \right\rangle_J,
\end{aligned}
\tag{46}
$$

where the normalisation property of the eigenvectors $1 = \sum_{i=1}^{N} u_{i\alpha}^2$ for any $\alpha = 1, \ldots, N$ and the Sokhotski-Plemelj identity (7) have been used. Given the connection with the eigenvectors, the resolvent allows for the study of localisation properties (see for instance [15, 23] and Section 7 for a further application).

The cavity method can be directly applied to the resolvent, as originally suggested in [42]. A set of self-consistency equations for the cavity diagonal entries of the resolvent $G(z)_{jj}^{(i)}$ (i.e. the diagonal entries of the resolvent matrix from which the $i$-th row and the $i$-th column have been removed) can be obtained thanks to a Schur decomposition procedure or alternatively by representing the $G(z)_{ii}$ and in turn the $G(z)_{jj}^{(i)}$ as Gaussian integrals and then applying the cavity method in the same fashion as Section 3.2 (see e.g. [23] for the details).

A correspondence between the cavity formalism as developed in Section 3.2 and the cavity method applied to the resolvent can be easily established. Indeed, comparing respectively Eq. (27) and (21) with Eq. (16) and (13) in [23], it follows that for any $\lambda \in \mathbb{R}$ and for any $i, j = 1, \ldots, N$

$$G(\lambda - i\varepsilon)_{jj}^{(i)} = \frac{i}{\omega_j^{(i)}}, \tag{47}$$

$$G(\lambda - i\varepsilon)_{ii} = \frac{i}{\omega_i}, \tag{48}$$

where the $\omega_j^{(i)}$ and the $\omega_i$ are defined respectively in Eq. (29) and (31).

# 4 Replica method: the Bray-Rodgers equation

In this section, we illustrate the replica calculation for the average spectral density, as originally proposed by Bray and Rodgers. Following [28], the goal is to evaluate the average spectral density (1) of an ensemble of $N \times N$ real symmetric sparse matrices. Leveraging on the notation of Section 3, given a matrix $J$, each matrix entry can be written as $J_{ij} = c_{ij} K_{ij}$, where the $c_{ij} = \{0, 1\}$ represent the pure adjacency matrix of the underlying graph and the $K_{ij}$ encode the bond weights. In particular, the matrix model considered in [28] is the Erdős-Rényi (ER) model: the probability of having a non-zero entry is given by $p = c/N$, where $c$ represents the mean degree of the nodes of the underlying graph. For more details on the ER model, see Appendix C. The joint distribution of the matrix entries is given by

$$P(\{J_{ij}\}) = \prod_{i<j} p_C(c_{ij}) \delta_{c_{ij}, c_{ji}} \prod_{i<j} p_K(K_{ij}) \delta_{K_{ij}, K_{ji}}, \tag{49}$$

where $p_C(c_{ij})$ represents the ER connectivity distribution, viz.

$$p_C(c_{ij}) = \frac{c}{N} \delta_{c_{ij}, 1} + \left(1 - \frac{c}{N}\right) \delta_{c_{ij}, 0}, \tag{50}$$

while $p_K(K_{ij})$ represents the bond weight pdf, which will be kept unspecified until the very end of the calculation.

## 4.1 Replica derivation

The Edwards-Jones formula (2) is used. As anticipated in Section 2, in order to deal with the *quenched* average (16) the replica identity will be employed, viz.

$$\langle \log Z(\lambda) \rangle_J = \lim_{n \to 0} \frac{1}{n} \log \langle Z(\lambda)^n \rangle_J, \tag{51}$$

where $n$ is initially taken as an integer [3]. The replica identity is easily obtained considering that in the limit $n \to 0$

$$\log \langle Z(\lambda)^n \rangle_J = \log \left(1 + n \langle \log Z(\lambda) \rangle_J + \mathcal{O}(n^2)\right) \simeq n \langle \log Z(\lambda) \rangle_J. \tag{52}$$

The average replicated version of the partition function (3) reads

$$\langle Z(\lambda)^n \rangle_J = \int \prod_{a=1}^n \prod_{i=1}^N dv_{ia} \exp\left(-\frac{i}{2} \lambda_\varepsilon \sum_{i=1}^N \sum_{a=1}^n v_{ia}^2\right) \left\langle \exp\left(\frac{i}{2} \sum_{i,j=1}^N \sum_{a=1}^n v_{ia} J_{ij} v_{ja}\right) \right\rangle_J. \tag{53}$$

The ensemble average $\langle ... \rangle_J$ splits into the connectivity average w.r.t. the $c_{ij}$ and the disorder average w.r.t. the $K_{ij}$. The connectivity average can be performed explicitly exploiting the large $N$ scaling, yielding

$$\left\langle \exp\left(\frac{i}{2} \sum_{i,j=1}^N \sum_{a=1}^n v_{ia} J_{ij} v_{ja}\right) \right\rangle_J = \exp\left[\frac{c}{2N} \sum_{i,j=1}^N \left(\left\langle e^{iK \sum_a v_{ia} v_{ja}}\right\rangle_K - 1\right)\right], \tag{54}$$

where $\langle ... \rangle_K$ denotes averaging over $p_K(K)$. The details of the calculation leading to (54) can be found in D.

---

[3]It is implicitly expected that the average replicated partition function $\langle Z(\lambda)^n \rangle_J$ could be analytically continued in the vicinity of $n = 0$ in a safe manner, although in principle this is not guaranteed.

In order to decouple sites, the following functional order parameter is introduced,

$$\varphi(\vec{v}) = \frac{1}{N}\sum_{i=1}^{N}\prod_{a=1}^{n}\delta(v_a - v_{ia}),\tag{55}$$

via the path integral identity

$$1 = N\int\mathcal{D}\varphi\mathcal{D}\hat{\varphi}\exp\left[-\mathrm{i}\int\mathrm{d}\vec{v}\,\hat{\varphi}(\vec{v})\left(N\varphi(\vec{v})-\sum_{i=1}^{N}\prod_{a=1}^{n}\delta(v_a - v_{ia})\right)\right],\tag{56}$$

where $\vec{v}\in\mathbb{R}^n$ represents a $n$-dimensional vector in the replica space. Eq. (56) is the functional analogue of

$$1 = \int\mathrm{d}x\,\delta(x-\bar{x}) = \int\mathrm{d}x\int\frac{\mathrm{d}k}{2\pi}\mathrm{e}^{-\mathrm{i}k(x-\bar{x})}.\tag{57}$$

In terms of the order parameter (55), the average replicated partition function becomes

$$\langle Z(\lambda)^n\rangle_J = N\int\mathcal{D}\varphi\mathcal{D}\hat{\varphi}\exp\left(-\mathrm{i}N\int\mathrm{d}\vec{v}\,\hat{\varphi}(\vec{v})\varphi(\vec{v})\right)$$

$$\times\exp\left[\frac{Nc}{2}\int\mathrm{d}\vec{v}\,\mathrm{d}\vec{v'}\,\varphi(\vec{v})\varphi(\vec{v'})\left(\left\langle\mathrm{e}^{\mathrm{i}K\sum_a v_a v'_a}\right\rangle_K - 1\right)\right]$$

$$\times\int\prod_{a=1}^{n}\prod_{i=1}^{N}\mathrm{d}v_{ia}\exp\left[-\frac{\mathrm{i}}{2}\lambda_\varepsilon\sum_{i=1}^{N}\sum_{a=1}^{n}v_{ia}^2 + \mathrm{i}\sum_{i=1}^{N}\int\mathrm{d}\vec{v}\,\hat{\varphi}(\vec{v})\prod_{a=1}^{n}\delta(v_a - v_{ia})\right].\tag{58}$$

The multiple integral $I$ in the last line of (58) factorises into $N$ identical copies of the same $n$-dimensional integral over $\mathbb{R}^n$. Indeed, one finds

$$I = \int\prod_{a=1}^{n}\prod_{i=1}^{N}\mathrm{d}v_{ia}\exp\left(-\frac{\mathrm{i}}{2}\lambda_\varepsilon\sum_{a=1}^{n}\sum_{i=1}^{N}v_{ia}^2 + \mathrm{i}\sum_{i=1}^{N}\hat{\varphi}(\vec{v}_i)\right)$$

$$= \left[\int\prod_{a=1}^{n}\mathrm{d}v_a\exp\left(-\frac{\mathrm{i}}{2}\lambda_\varepsilon\sum_{a=1}^{n}v_a^2 + \mathrm{i}\hat{\varphi}(\vec{v})\right)\right]^N$$

$$= \exp\left[N\mathrm{Log}\int\mathrm{d}\vec{v}\exp\left(-\frac{\mathrm{i}}{2}\lambda_\varepsilon\sum_{a=1}^{n}v_a^2 + \mathrm{i}\hat{\varphi}(\vec{v})\right)\right],\tag{59}$$

where Log denotes again the principal branch of the complex logarithm.

The replicated partition function (58) can then be written in the form

$$\langle Z(\lambda)^n\rangle_J \propto \int\mathcal{D}\varphi\mathcal{D}\hat{\varphi}\exp\left(NS_n[\varphi,\hat{\varphi},\lambda]\right),\tag{60}$$

where

$$S_n[\varphi,\hat{\varphi},\lambda] = S_1[\varphi,\hat{\varphi}] + S_2[\varphi] + S_3[\hat{\varphi},\lambda],\tag{61}$$

with

$$S_1[\varphi,\hat{\varphi}] = -\mathrm{i}\int\mathrm{d}\vec{v}\,\hat{\varphi}(\vec{v})\varphi(\vec{v}),\tag{62}$$

$$S_2[\varphi] = \frac{c}{2}\int\mathrm{d}\vec{v}\,\mathrm{d}\vec{v'}\,\varphi(\vec{v})\varphi(\vec{v'})\left(\left\langle\mathrm{e}^{\mathrm{i}K\sum_a v_a v'_a}\right\rangle_K - 1\right),\tag{63}$$

$$S_3[\hat{\varphi},\lambda] = \mathrm{Log}\int\mathrm{d}\vec{v}\exp\left(-\frac{\mathrm{i}}{2}\lambda_\varepsilon\sum_{a=1}^{n}v_a^2 + \mathrm{i}\hat{\varphi}(\vec{v})\right).\tag{64}$$

Eq. (60) is amenable to a saddle-point evaluation for large $N$, yielding

$$\langle Z(\lambda)^n \rangle_J \approx \exp\left(N S_n[\varphi^\star, \hat{\varphi}^\star, \lambda]\right), \tag{65}$$

where the star denotes the saddle-point value of the order parameter and its conjugate. The stationarity conditions of the action (61) w.r.t. the functional order parameter $\varphi$ and its conjugate $\hat{\varphi}$ give

$$\left.\frac{\delta S_n}{\delta \varphi}\right|_{\varphi^\star, \hat{\varphi}^\star} = 0 \Rightarrow i\hat{\varphi}^\star(\vec{v}) = c \int d\vec{v}' \varphi^\star(\vec{v}') \left[\left\langle \exp\left(iK \sum_a v_a v_a'\right)\right\rangle_K - 1\right], \tag{66}$$

$$\left.\frac{\delta S_n}{\delta \hat{\varphi}}\right|_{\varphi^\star, \hat{\varphi}^\star} = 0 \Rightarrow \varphi^\star(\vec{v}) = \frac{\exp\left[-\frac{i}{2}\lambda_\varepsilon \sum_a v_a^2 + i\hat{\varphi}^\star(\vec{v})\right]}{\int d\vec{v}' \exp\left[-\frac{i}{2}\lambda_\varepsilon \sum_a v_a'^2 + i\hat{\varphi}^\star(\vec{v}')\right]}. \tag{67}$$

The two stationarity conditions (66) and (67) can be combined. Indeed, by calling $i\hat{\varphi}^\star(\vec{v}) = c g(\vec{v})$ and inserting (67) in (66), one obtains

$$g(\vec{v}) = \frac{\int d\vec{v}' f(\vec{v} \cdot \vec{v}') \exp\left[-\frac{i}{2}\lambda_\varepsilon \sum_a v_a'^2 + c g(\vec{v}')\right]}{\int d\vec{v}' \exp\left[-\frac{i}{2}\lambda_\varepsilon \sum_a v_a'^2 + c g(\vec{v}')\right]}, \tag{68}$$

where $f(x) = \langle e^{iKx} \rangle_K - 1$. A numerical solution for coupled saddle-point equations that are analogous to Eq. (66) and (67) has been recently proposed in [68].

## 4.2 Average spectral density: replica symmetry assumption

The function $g(\vec{v})$ defined by (68) fully determines the average spectral density. Indeed, recalling (2) and using (65), one gets

$$\begin{aligned}
\rho(\lambda) &= -\frac{2}{\pi N} \lim_{\varepsilon \to 0^+} \text{Im} \frac{\partial}{\partial \lambda} \langle \log Z(\lambda) \rangle_J \\
&= -\frac{2}{\pi N} \lim_{\varepsilon \to 0^+} \text{Im} \frac{\partial}{\partial \lambda} \lim_{n \to 0} \frac{1}{n} \log \langle Z^n(\lambda) \rangle_J \\
&\approx -\frac{2}{\pi N} \lim_{\varepsilon \to 0^+} \text{Im} \frac{\partial}{\partial \lambda} \lim_{n \to 0} \frac{1}{n} \log \left[\exp\left(N S_n[\varphi^\star, \hat{\varphi}^\star, \lambda]\right)\right] \\
&= -\frac{2}{\pi} \lim_{\varepsilon \to 0^+} \text{Im} \lim_{n \to 0} \frac{1}{n} \frac{\partial}{\partial \lambda} S_n[\varphi^\star, \hat{\varphi}^\star, \lambda].
\end{aligned} \tag{69}$$

The $\lambda$-derivative acts only on the terms of the action $S_n$ explicitly depending on $\lambda$, due to the stationarity of $S_n$ w.r.t. $\varphi^\star$ and $\hat{\varphi}^\star$. Indeed, one obtains

$$\frac{\partial}{\partial \lambda} S_n[\varphi^\star, \hat{\varphi}^\star, \lambda] = \frac{\partial \varphi}{\partial \lambda} \left.\frac{\delta S_n}{\delta \varphi}\right|_{\varphi = \varphi^\star, \hat{\varphi} = \hat{\varphi}^\star} + \frac{\partial \hat{\varphi}}{\partial \lambda} \left.\frac{\delta S_n}{\delta \hat{\varphi}}\right|_{\varphi = \varphi^\star, \hat{\varphi} = \hat{\varphi}^\star} + \left.\frac{\partial S_n}{\partial \lambda}\right|_{\varphi = \varphi^\star, \hat{\varphi} = \hat{\varphi}^\star} = \frac{\partial S_3[\hat{\varphi}^\star, \lambda]}{\partial \lambda}, \tag{70}$$

with $S_3[\hat{\varphi}^\star, \lambda]$ defined in (64) entailing the ratio

$$\frac{\partial}{\partial \lambda} S_n[\varphi^\star, \hat{\varphi}^\star, \lambda] = \frac{-\frac{i}{2} \int d\vec{v} \left(\sum_a v_a^2\right) \exp\left[-\frac{i\lambda_\varepsilon}{2} \sum_a v_a^2 + c g(\vec{v})\right]}{\int d\vec{v} \exp\left[-\frac{i\lambda_\varepsilon}{2} \sum_a v_a^2 + c g(\vec{v})\right]}, \tag{71}$$

where $g(\vec{v})$ solves (68).

In order to perform the $n \to 0$ limit in (69), an assumption on the symmetries of the function $g(\vec{v})$, or equivalently of both $\varphi^\star(\vec{v})$ and $\hat{\varphi}^\star(\vec{v})$, under permutations of replica indices needs to be made. It is known that a replica-symmetric "high-temperature" solution, preserving

both permutational and rotational symmetry in the replica space, is exact in the random matrix context [4]. Hence, following [28], one can assume

$$g(\vec{v}) = g(v),\tag{72}$$

where $v = |\vec{v}| = \sqrt{\sum_a v_a^2}$. Therefore, taking into account the ratio (71), the replica symmetric ansatz (72) and that $\text{Im}(ix) = \text{Re}(x)$ for any $x \in \mathbb{C}$, the average spectral density reads

$$\rho(\lambda) = \frac{1}{\pi} \lim_{\varepsilon \to 0^+} \text{Re} \lim_{n \to 0} \frac{1}{n} \frac{\int d\vec{v} \, v^2 \exp\left[-\frac{i\lambda_\varepsilon}{2} v^2 + c g(v)\right]}{\int d\vec{v} \exp\left[-\frac{i\lambda_\varepsilon}{2} v^2 + c g(v)\right]} \, .\tag{73}$$

To further simplify the ratio of integrals in (73), $n$-dimensional spherical coordinates are introduced. The symbol $v$ represents the radial coordinate and $\vec{\phi} = \{\phi_1, \phi_2, ..., \phi_{n-1}\}$ are $n-1$ angular coordinates, with $\phi_{n-1} \in [0, 2\pi]$ and $\phi_i \in [0, \pi]$ for $i = 1, ..., n-2$. The Jacobian of the coordinate transformation is $J(v, \phi_1, ..., \phi_{n-2}) = v^{n-1}(\sin(\phi_1))^{n-2}(\sin(\phi_2))^{n-3} \cdots \sin(\phi_{n-2})$. In these coordinates, the factor arising from the integration over the angular degrees of freedom,

$$I(\vec{\phi}) = \int_{[0,\pi]^{n-2}} d\phi_1 d\phi_2 \cdots d\phi_{n-2} \int_0^{2\pi} d\phi_{n-1} (\sin(\phi_1))^{n-2} (\sin(\phi_2))^{n-3} \cdots \sin(\phi_{n-2}), \quad (74)$$

appears both in the numerator and denominator of (73). Hence it cancels, yielding eventually

$$\rho(\lambda) = \frac{1}{\pi} \lim_{\varepsilon \to 0^+} \text{Re} \lim_{n \to 0} \frac{1}{n} \frac{\int_0^\infty dv \, v^{n+1} \exp\left[-\frac{i\lambda_\varepsilon}{2} v^2 + c g(v)\right]}{\int_0^\infty dv \, v^{n-1} \exp\left[-\frac{i\lambda_\varepsilon}{2} v^2 + c g(v)\right]} \, .\tag{75}$$

The integral in the denominator can be further simplified integrating by parts, i.e.

$$\int_0^\infty dv \, v^{n-1} \exp\left[-\frac{i\lambda_\varepsilon}{2} v^2 + c g(v)\right] = \frac{1}{n} \int_0^\infty dv \, v^n \exp\left[-\frac{i\lambda_\varepsilon}{2} v^2 + c g(v)\right] \left(i\lambda_\varepsilon v - c g'(v)\right),\tag{76}$$

since the boundary contribution vanishes. Therefore, taking the $n \to 0$ limit, the average spectral density reads

$$\rho(\lambda) = \frac{1}{\pi} \lim_{\varepsilon \to 0^+} \text{Re} \frac{\int_0^\infty dv \, v \exp\left[-\frac{i\lambda_\varepsilon}{2} v^2 + c g(v)\right]}{\int_0^\infty dv \exp\left[-\frac{i\lambda_\varepsilon}{2} v^2 + c g(v)\right] (i\lambda_\varepsilon v - c g'(v))} \, .\tag{77}$$

The expression (77) shows that the function $g(v)$ is the only ingredient needed to compute the average spectral density. The search for a (replica-symmetric) solution of (68) will be addressed in Section 4.3.

## 4.3 Replica symmetry assumption for the Bray-Rodgers integral equation

At this stage, in order to get a replica-symmetric version of (68), the replica-symmetric ansatz (72) is applied and spherical coordinates are again introduced. Assuming that $\phi_1 = \phi \in [0, \pi]$ is the angle between the vectors $\vec{v}$ and $\vec{v'}$ and $|\vec{v'}| = r$ one finds

$$g(v) = \frac{\int_0^\infty dr \, r^{n-1} \exp\left[-\frac{1}{2}\lambda_\varepsilon r^2 + c g(r)\right] \int_0^\pi d\phi (\sin\phi)^{n-2} f(vr\cos\phi)}{\int_0^\infty dr \, r^{n-1} \exp\left[-\frac{1}{2}\lambda_\varepsilon r^2 + c g(r)\right] \int_0^\pi d\phi (\sin\phi)^{n-2}} \, ,\tag{78}$$

---

[4]Rotational invariance in replica space is a stronger condition than the symmetry upon permutation of replicas. An example of an ansatz satisfying permutational symmetry, but not rotational invariance would be $g(\vec{v}) = g(\sum_{a=1}^n v_a)$.

since all the remaining angular integrations cancel between numerator and denominator. We recall that $f(z) = \langle e^{iKz} \rangle_K - 1$. In order to proceed, a specific bond weight distribution must be introduced. The choice made by Rodgers and Bray in [28],

$$p_K(K) = \frac{1}{2}\delta_{K,1} + \frac{1}{2}\delta_{K,-1}, \tag{79}$$

entails that $f(z) = \cos z - 1$. This gives

$$g(v) = \frac{\int_0^\infty dr\, r^{n-1} \exp\left[-\frac{1}{2}\lambda_\varepsilon r^2 + cg(r)\right] \int_0^\pi d\phi\, (\sin\phi)^{n-2}[\cos(vr\cos\phi) - 1]}{\int_0^\infty dr\, r^{n-1} \exp\left[-\frac{i}{2}\lambda_\varepsilon r^2 + cg(r)\right] \int_0^\pi d\phi\, (\sin\phi)^{n-2}}. \tag{80}$$

The angular integral in the numerator in (80) yields (see formula 21 of Section 3.715 in [69])

$$\begin{aligned}
I_{\text{num}}^{\text{ang}} &= \int_0^\pi d\phi\, (\sin\phi)^{n-2}[\cos(vr\cos\phi) - 1] \\
&= \frac{\sqrt{\pi}\Gamma\left(\frac{n-1}{2}\right)}{2}\left[\left(\frac{2}{vr}\right)^{\frac{n}{2}}\left(nJ_{\frac{n}{2}}(vr) - vrJ_{\frac{n}{2}+1}(vr)\right) - \frac{2}{\Gamma\left(\frac{n}{2}\right)}\right],
\end{aligned} \tag{81}$$

where $J_\alpha(x)$ indicates the Bessel function of the first kind, defined by the series

$$J_\alpha(x) = \sum_{m=0}^\infty \frac{(-1)^m}{m!\Gamma(m+\alpha+1)}\left(\frac{x}{2}\right)^{2m+\alpha}. \tag{82}$$

The angular integral in the denominator of (80) is independent of the radial one and gives

$$I_{\text{den}}^{\text{ang}} = \int_0^\pi d\phi\, (\sin\phi)^{n-2} = \frac{\sqrt{\pi}\Gamma\left(\frac{n-1}{2}\right)}{\Gamma\left(\frac{n}{2}\right)}, \tag{83}$$

thus canceling the divergent factor for $n < 1$ appearing in (81). The radial integral in the denominator in (80) can be simplified integrating by parts. By calling $G(r) = \exp\left[-\frac{i}{2}\lambda_\varepsilon r^2 + cg(r)\right]$, one obtains

$$I_{\text{den}}^{\text{rad}} = \int_0^\infty dr\, r^{n-1}G(r) = -\frac{1}{n}\int_0^\infty dr\, r^n G'(r), \tag{84}$$

as the boundary contribution vanishes. Collecting the results, one finds

$$g(v) = -\frac{n\Gamma\left(\frac{n}{2}\right)}{2}\frac{\int_0^\infty dr\, r^{n-1}G(r)\left[\left(\frac{2}{vr}\right)^{\frac{n}{2}}\left(nJ_{\frac{n}{2}}(vr) - vrJ_{\frac{n}{2}+1}(vr)\right) - \frac{2}{\Gamma\left(\frac{n}{2}\right)}\right]}{\int_0^\infty dr\, r^n G'(r)}. \tag{85}$$

Lastly, the $n \to 0$ limit in (85) is taken. Recalling the definition of $G(r)$ and noticing that

1.
$$\Gamma(n) \approx \frac{1}{n} \quad \text{as} \quad n \to 0 \Rightarrow \lim_{n \to 0} -n\Gamma\left(\frac{n}{2}\right) = -2, \tag{86}$$

2.
$$\lim_{n \to 0}\int_0^\infty dr\, r^n G'(r) = \int_0^\infty dr\, G'(r) = G(\infty) - G(0) = -e^{cg(0)}, \tag{87}$$

3.
$$\lim_{n \to 0} r^{n-1}G(r)\left[\left(\frac{2}{vr}\right)^{\frac{n}{2}}\left(nJ_{\frac{n}{2}}(vr) - vrJ_{\frac{n}{2}+1}(vr)\right) - \frac{2}{\Gamma\left(\frac{n}{2}\right)}\right] = -vJ_1(vr)G(r), \tag{88}$$

one obtains

$$g(v) = \frac{v \int_0^\infty \mathrm{d}r J_1(vr) \exp\left[-\frac{\mathrm{i}}{2}\lambda_\varepsilon r^2 + cg(r)\right]}{-\mathrm{e}^{cg(0)}}. \tag{89}$$

Given the structure of Eq. (89), it follows that $g(0) = 0$, therefore eventually

$$g(v) = -v \int_0^\infty \mathrm{d}r J_1(vr) \exp\left[-\frac{\mathrm{i}}{2}\lambda_\varepsilon r^2 + cg(r)\right], \tag{90}$$

which is equivalent to Eq. (18) in [28]. This equation, also known as the Bray-Rodgers integral equation, fully defines the quantity $g(v)$. Despite numerous attempts, an exact analytical solution for (90) is currently not available. The numerical evaluation of Eq. (90) in the case of $c = 20$ has been carried out in [17], to obtain the average spectral density of Laplacians of ER graphs with bimodal weights. In this high connectivity regime, the quality of the numerical solution was comparable with the SDA approximation (see [33]). On the other hand, the authors in [16] describe a procedure for the solution of a similar integral equation employing a series expansion which is valid only for $c < 1/2$. Nonetheless, for values of $c$ that are in between these two extremal cases, Eq. (90) has proved to be hard to tackle even numerically, due to the exponential non-linearity and the oscillatory Bessel term.

Taking into account Eq. (90), the average spectral density (77) can be further simplified as follows. Indeed, recalling that $G(v) = \exp\left[-\mathrm{i}\frac{\lambda_\varepsilon}{2}v^2 + cg(v)\right]$, the denominator in Eq. (77) can be expressed as

$$I_{\rho,\mathrm{den}} = -\int_0^\infty \mathrm{d}v \, G'(v) = G(0) - G(\infty) = \mathrm{e}^{cg(0)} = 1, \tag{91}$$

therefore entailing that the average spectral density reduces to

$$\rho(\lambda) = \frac{1}{\pi} \lim_{\varepsilon \to 0^+} \mathrm{Re} \int_0^\infty \mathrm{d}v \, v \exp\left[-\frac{\mathrm{i}\lambda_\varepsilon}{2}v^2 + cg(v)\right]. \tag{92}$$

### 4.4 The average spectral density in the $c \to \infty$ limit

It can be shown that Eq. (90) can be solved perturbatively in powers of $1/c$ in the limit $c \to \infty$. In this framework, the average spectral density (92) is in turn expressed as a perturbative expansion, whose leading term is the Wigner semicircular law. We will provide a derivation inspired by [28, 30].

First, the following changes of variables are introduced,

$$v^2 = -\frac{2\mathrm{i}s}{\lambda_\varepsilon}, \tag{93}$$

$$r^2 = -\frac{2\mathrm{i}u}{\lambda_\varepsilon}, \tag{94}$$

$$\lambda_\varepsilon^2 = cx_\delta^2, \tag{95}$$

where $x_\delta = x - \mathrm{i}\delta$, with $\delta > 0$. Moreover, it is assumed that $g(v) = \frac{1}{c}\gamma(s)$.

Eq. (95) implies rescaling the spectral density such that a meaningful $c \to \infty$ limit can be taken. This rescaling is equivalent to normalising the matrix entries by $\sqrt{c}$. After the change of variables, the spectral density will be expressed in terms of $x_\delta = x - \mathrm{i}\delta$, which does not scale with $c$. In this setting, the limit $\varepsilon \to 0^+$ is replaced by the limit $\delta \to 0^+$. Taking into account (95), for the l.h.s. of Eq. (92) before taking the $\varepsilon \to 0^+$ limit one finds

$$\rho(\lambda_\varepsilon) = \rho(x_\delta)\frac{\mathrm{d}x_\delta}{\mathrm{d}\lambda_\varepsilon} = \rho(x_\delta)\frac{1}{\sqrt{c}}, \tag{96}$$

entailing that

$$\rho(\lambda) = \lim_{\varepsilon \to 0^+} \rho(\lambda_\varepsilon) = \lim_{\delta \to 0^+} \rho(x_\delta) \frac{1}{\sqrt{c}} = \rho(x) \frac{1}{\sqrt{c}} \ . \tag{97}$$

One then rewrites Eq. (90) in terms of the new variables. The differential in (90) transforms as

$$\mathrm{d}r = -\frac{\mathrm{i}}{\lambda_\varepsilon} \sqrt{-\frac{\lambda_\varepsilon}{2\mathrm{i}u}} \mathrm{d}u = -\frac{\mathrm{i}}{\sqrt{c}x_\delta} \sqrt{-\frac{\sqrt{c}x_\delta}{2\mathrm{i}u}} \mathrm{d}u \ , \tag{98}$$

while the integration boundaries are unchanged. Indeed, from (94), one finds

$$u = \frac{\sqrt{c}r^2\delta}{2} + \mathrm{i}\frac{\sqrt{c}r^2 x}{2} = \frac{\sqrt{c}r^2}{2} \sqrt{x^2 + \delta^2} e^{\mathrm{i}\arctan\left(\frac{x}{\delta}\right)} = \begin{cases} 0 & r = 0 \\ \infty & r \to \infty \end{cases} \ . \tag{99}$$

In terms of the new variables, after some algebra Eq. (90) converts to

$$\gamma(s) = \frac{s}{x_\delta^2} \int_0^\infty \mathrm{d}u \exp\left[-u + \gamma(u)\right] \sum_{m=0}^\infty \frac{1}{m!(m+1)!} \left(\frac{su}{cx_\delta^2}\right)^m , \tag{100}$$

corresponding to Eq. (23) in [28].

With these choices, it is natural to expand $\gamma(s)$ as a power series in $s/c$. High powers of $s$ are related to high powers of $1/c$. Therefore, one expects to find a solution of the form

$$\gamma(s) = c \sum_{r=1}^\infty b_r \left(\frac{s}{c}\right)^r , \tag{101}$$

where in turn the coefficients $b_r$ are defined via the expansion

$$b_r = \sum_{\ell=0}^\infty \frac{b_r^{(\ell)}}{c^\ell} \ . \tag{102}$$

Eq. (101) and (102) allow one to obtain all possible combinations of powers of $\frac{s^r}{c^{r+\ell}}$. The target is to determine the coefficients $b_r^{(\ell)}$, by solving Eq. (100) order by order. This would permit a complete representation of $\gamma(s)$ as a power series. The following steps will be followed for the solution.

- Express $\gamma$ via the expansions (101) and (102) in both the l.h.s. and the exponent of the r.h.s. of (100).

- Integrate w.r.t. $u$ term by term in the r.h.s. of (100).

- Equate the coefficients of the powers $\frac{s^r}{c^{r+\ell}}$.

The expansion of $\gamma(s)$ will be stopped at $\mathcal{O}\left(\frac{1}{c}\right)$. Hence, the l.h.s. of (100) reads

$$\gamma(s) = b_1 s + b_2 \frac{s^2}{c} + \mathcal{O}\left(\frac{s^2}{c^2}\right) = b_1^{(0)} s + b_1^{(1)} \frac{s}{c} + b_2^{(0)} \frac{s^2}{c} + \mathcal{O}\left(\frac{s^2}{c^2}\right) . \tag{103}$$

Looking at the r.h.s., one notices that only the terms $m = 0$ and $m = 1$ of the sum in (100) are needed to match the powers $\frac{s^r}{c^{r+\ell}}$ in (103). Indeed, one finds

$$\gamma(s) = \frac{s}{x_\delta^2} \int_0^\infty \mathrm{d}u \exp\left[-u + \gamma(u)\right] + \frac{s^2}{2cx_\delta^4} \int_0^\infty \mathrm{d}u \, u \exp\left[-u + \gamma(u)\right] + \mathcal{O}\left(\frac{s^2}{c^2}\right) . \tag{104}$$

The first and second integral on the r.h.s. of (104) can be denoted respectively as $I_A$ and $I_B$. Considering first the integral $I_A$, it has a pre-factor of order $\mathcal{O}(s)$, therefore it may yield contributions of any order in powers of $1/c$ depending on the order at which we stop the expansion of $\gamma(u)$ in the exponent. Expanding $\gamma(u)$ as in (103) is sufficient to obtain all the $\mathcal{O}(1)$ and $\mathcal{O}\left(\frac{1}{c}\right)$ contributions. Indeed, we have

$$
\begin{aligned}
I_A &= \int_0^\infty du \exp\left[-u + \gamma(u)\right] \\
&\simeq \int_0^\infty du \exp\left[\frac{b_2^{(0)}}{c}u^2 + \left(b_1^{(0)} - 1 + \frac{b_1^{(1)}}{c}\right)u\right] \\
&= \frac{\sqrt{\pi}}{2\sqrt{-\frac{b_2^{(0)}}{c}}}e^{y^2}\operatorname{erfc}(y),
\end{aligned}
\tag{105}
$$

where

$$
y = -\frac{b_1^{(0)} - 1 + \frac{b_1^{(1)}}{c}}{2\sqrt{-\frac{b_2^{(0)}}{c}}},
\tag{106}
$$

and $\operatorname{Re}\left(\frac{b_2^{(0)}}{c}\right) < 0$. The function denoted by $\operatorname{erfc}(z)$ is the complementary error function, defined for real $z$ as

$$
\operatorname{erfc}(z) = \frac{2}{\sqrt{\pi}}\int_z^\infty dt\ e^{-t^2}.
\tag{107}
$$

In order to express (105) as a power series, an asymptotic expansion of $\operatorname{erfc}(z)$ is employed. For large real $z$ it is given by

$$
\operatorname{erfc}(z) \simeq \frac{e^{-z^2}}{z\sqrt{\pi}}\left[1 + \sum_{n=1}^\infty (-1)^n \frac{(2n)!}{n!(2z)^{2n}}\right].
\tag{108}
$$

The series is divergent for any finite $z$. However, few terms of it are sufficient to approximate $\operatorname{erfc}(z)$ well for any finite $z$. Using (108) in (105), one obtains

$$
\begin{aligned}
I_A &\approx -\frac{1}{b_1^{(0)} - 1 + \frac{b_1^{(1)}}{c}}\left[1 + \sum_{n=1}^\infty \frac{(2n)!\left(\frac{b_2^{(0)}}{c}\right)^n}{n!\left(b_1^{(0)} - 1 + \frac{b_1^{(1)}}{c}\right)^{2n}}\right] \\
&= -\frac{1}{b_1^{(0)} - 1 + \frac{b_1^{(1)}}{c}} - \sum_{n=1}^\infty \frac{(2n)!}{n!}\left(\frac{b_2^{(0)}}{c}\right)^n \frac{1}{\left(b_1^{(0)} - 1 + \frac{b_1^{(1)}}{c}\right)^{2n+1}}.
\end{aligned}
\tag{109}
$$

Eq. (109) can be further simplified recalling that $(1 + \beta x)^\alpha \approx 1 + \alpha\beta x$ for $x \ll 1$, entailing that the first term of (109) becomes

$$
-\frac{1}{b_1^{(0)} - 1 + \frac{b_1^{(1)}}{c}} \approx \frac{1}{1 - b_1^{(0)}} + \frac{b_1^{(1)}}{\left(1 - b_1^{(0)}\right)^2}\frac{1}{c}.
\tag{110}
$$

Since the expansion is stopped at $\mathcal{O}\left(\frac{1}{c}\right)$, only the $n = 1$ term in the sum in (109) needs to be considered, as the contributions for $n \geq 2$ are at least $\mathcal{O}\left(\frac{1}{c^2}\right)$. Moreover, since the $n = 1$ term exhibits the $\mathcal{O}\left(\frac{1}{c}\right)$ scaling explicitly, only the $\mathcal{O}(1)$ contribution arising from the round brackets

in the denominator of the general term of the sum in (109) must be taken into account. Indeed, using (110) the $n = 1$ term in (109) becomes

$$2\frac{b_2^{(0)}}{c}\left(-\frac{1}{b_1^{(0)} - 1 + \frac{b_1^{(1)}}{c}}\right)^3 \approx 2\frac{b_2^{(0)}}{c}\left(\frac{1}{1 - b_1^{(0)}} + \frac{b_1^{(1)}}{\left(1 - b_1^{(0)}\right)^2}\frac{1}{c}\right)^3$$

$$= 2\frac{b_2^{(0)}}{\left(1 - b_1^{(0)}\right)^3}\frac{1}{c} + \mathcal{O}\left(\frac{1}{c^2}\right). \tag{111}$$

Collecting all the leading contributions to the integral $I_A$ one obtains

$$I_A = \int_0^\infty du \exp[-u + \gamma(u)] = \frac{1}{1 - b_1^{(0)}} + \frac{1}{c}\left[\frac{b_1^{(1)}}{\left(1 - b_1^{(0)}\right)^2} + \frac{2b_2^{(0)}}{\left(1 - b_1^{(0)}\right)^3}\right] + \mathcal{O}\left(\frac{1}{c^2}\right). \tag{112}$$

Considering now the second integral on the r.h.s. of (104), denoted by $I_B$, its pre-factor has already the $\mathcal{O}\left(\frac{1}{c}\right)$ scaling. Therefore, only the $\mathcal{O}(1)$ term arising from the $\frac{1}{c}$ expansion of $I_B$ is needed. To this purpose, it is sufficient to consider the expansion of $\gamma(u)$ no further than $\mathcal{O}\left(\frac{u}{c}\right)$. Indeed, one finds

$$I_B = \int_0^\infty du\, u \exp[-u + \gamma(u)]$$

$$\approx \int_0^\infty du\, u \exp\left[-\left(1 - b_1^{(0)} - \frac{b_1^{(1)}}{c}\right)u\right]$$

$$\approx \frac{1}{\left(1 - b_1^{(0)}\right)^2} + \mathcal{O}\left(\frac{1}{c}\right), \tag{113}$$

with $\mathrm{Re}\left(1 - b_1^{(0)} - \frac{b_1^{(1)}}{c}\right) > 0$.

In conclusion, using the expansions of $I_A$ and $I_B$, respectively given by Eq. (112) and (113), Eq. (104) representing the r.h.s. of (100) becomes

$$\gamma(s) = \frac{1}{x_\delta^2\left(1 - b_1^{(0)}\right)}s + \frac{1}{x_\delta^2}\left[\frac{b_1^{(1)}}{\left(1 - b_1^{(0)}\right)^2} + 2\frac{b_2^{(0)}}{\left(1 - b_1^{(0)}\right)^3}\right]\frac{s}{c} + \frac{1}{2x_\delta^4}\frac{1}{\left(1 - b_1^{(0)}\right)^2}\frac{s^2}{c} + \mathcal{O}\left(\frac{s^2}{c^2}\right). \tag{114}$$

Equating term by term the expansion in Eq. (103) and (114), one gets a closed set of equations to determine the three coefficients $b_1^{(0)}$, $b_1^{(1)}$ and $b_2^{(0)}$, viz.

$$\mathcal{O}(s): \quad b_1^{(0)} = \frac{1}{x_\delta^2\left(1 - b_1^{(0)}\right)}, \tag{115}$$

$$\mathcal{O}\left(\frac{s}{c}\right): \quad b_1^{(1)} = \frac{1}{x_\delta^2}\left[\frac{b_1^{(1)}}{\left(1 - b_1^{(0)}\right)^2} + \frac{2b_2^{(0)}}{\left(1 - b_1^{(0)}\right)^3}\right], \tag{116}$$

$$\mathcal{O}\left(\frac{s^2}{c}\right): \quad b_2^{(0)} = \frac{1}{2x_\delta^4\left(1 - b_1^{(0)}\right)^2}, \tag{117}$$

corresponding to Eq. (25), (26) and (27) in [28]. The latter system of equations can be easily solved, yielding

$$b_1^{(0)} = \frac{1}{2}\left[1 \pm A^{\frac{1}{2}}\right],\qquad(118)$$

$$b_1^{(1)} = \mp \frac{x_\delta^2}{16}\frac{\left[1 \pm A^{\frac{1}{2}}\right]^4}{A^{\frac{1}{2}}},\qquad(119)$$

$$b_2^{(0)} = \frac{1}{2}\left(b_1^{(0)}\right)^2,\qquad(120)$$

where $A = 1 - \frac{4}{x_\delta^2}$.

Finally, the average spectral density (92) can be evaluated perturbatively. Indeed, taking into account Eq. (97) and applying the change of variables (93), (95) and $g(v) = \frac{1}{c}\gamma(s)$ to the r.h.s. of Eq. (92), one finds

$$\rho(x)\frac{1}{\sqrt{c}} = \frac{1}{\pi}\lim_{\delta\to0^+}\mathrm{Re}\left[\left(-\frac{i}{\sqrt{c}x_\delta}\right)\int_0^\infty ds\,\exp\left[-s+\gamma(s)\right]\right] \Rightarrow$$

$$\rho(x) = \frac{1}{\pi}\lim_{\delta\to0^+}\mathrm{Im}\left[\frac{1}{x-i\delta}\int_0^\infty ds\,\exp\left[-s+\gamma(s)\right]\right],\qquad(121)$$

where the dependence on $c$ is only through the power series (101) expressing $\gamma(s)$.

The goal is to obtain the $\mathcal{O}(1)$ leading term dominating in Eq. (121) for large $c$, along with its $\mathcal{O}(1/c)$ correction. It is worth noticing that the integral appearing on the r.h.s. of Eq. (121) has been already evaluated up to $\mathcal{O}(\frac{1}{c})$. Indeed, it corresponds to the integral $I_A$ of Eq. (112).

Therefore, using (112) in (121) one gets

$$\rho(x) = \rho_0(x) + \frac{1}{c}\rho_1(x) + \mathcal{O}\left(\frac{1}{c^2}\right),\qquad(122)$$

where

$$\rho_0(x) = \frac{1}{\pi}\lim_{\delta\to0^+}\mathrm{Im}\left[\frac{1}{x-i\delta}\frac{1}{1-b_1^{(0)}}\right],\qquad(123)$$

$$\rho_1(x) = \frac{1}{\pi}\lim_{\delta\to0^+}\mathrm{Im}\left[\frac{1}{x-i\delta}\left(\frac{b_1^{(1)}}{\left(1-b_1^{(0)}\right)^2}+\frac{2b_2^{(0)}}{\left(1-b_1^{(0)}\right)^3}\right)\right],\qquad(124)$$

and the coefficients $b_1^{(0)}$, $b_1^{(1)}$ and $b_2^{(0)}$ have been defined respectively in (118), (119) and (120).

The $\mathcal{O}(1)$ leading term $\rho_0(x)$ in Eq. (123) is simply obtained by observing that

$$\mathrm{Im}\left[\frac{1}{x-i\delta}\frac{1}{1-b_1^{(0)}}\right] = \frac{1}{2}\mathrm{Im}\left[x-i\delta\pm\sqrt{(x-i\delta)^2-4}\right].\qquad(125)$$

Using (125), considering the imaginary part and taking the $\delta\to0^+$ limit, the $\mathcal{O}(1)$ leading term in Eq. (122) becomes

$$\rho_0(x) = \begin{cases} \frac{1}{2\pi}\sqrt{4-x^2} & -2 < x < 2 \\ 0 & \text{elsewhere} \end{cases},\qquad(126)$$

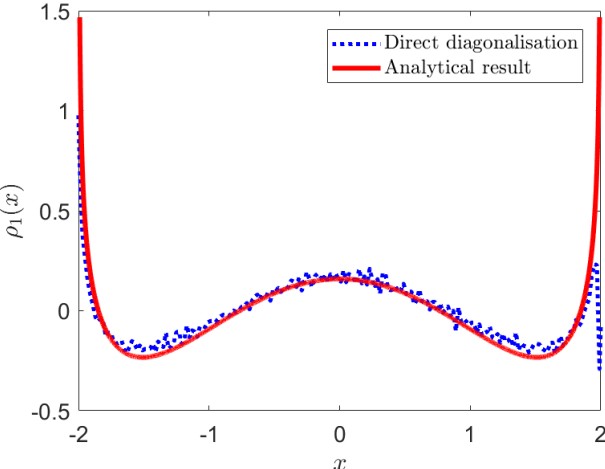

Figure 2: The $\mathcal{O}(1/c)$ correction $\rho_1(x)$ found in Eq. (128) to the average spectral density of ER matrices with bimodal weights distribution in the large $c$ limit (red line), compared to the histogram of $9.99 \times 10^6$ eigenvalues of matrices from the ER ensemble with $c = 50$ and bond weight distribution $p_K(K) = \frac{1}{2}\delta_{K,1/\sqrt{c}} + \frac{1}{2}\delta_{K,-1/\sqrt{c}}$ (blue dotted line). The $\mathcal{O}(1)$ contribution $\rho_0(x)$ in Eq. (126) has been removed from the histogram.

where we have used that

$$\text{Im}\left[\sqrt{y}\right] = \begin{cases} \sqrt{-y} & y < 0 \\ 0 & y > 0 \end{cases}, \tag{127}$$

and the plus sign in front of the square root has been chosen in order to get a physical (non-negative) solution. This sign choice amounts to selecting the top alternative for the signs appearing in the expansion coefficients (118), (119) and (120). Eq. (126) corresponds to the Wigner's semicircle as expected.

Similarly, using the definitions (118), (119) and (120), taking the $\delta \to 0^+$ limit and employing the property (127), after some algebra the $\mathcal{O}(1/c)$ correction (124) is obtained as

$$\rho_1(x) = \begin{cases} \frac{x^4 - 4x^2 + 2}{2\pi\sqrt{4 - x^2}} & -2 < x < 2 \\ 0 & \text{elsewhere} \end{cases}, \tag{128}$$

where the sign is determined by the same sign convention for the coefficients of the expansion adopted for the evaluation of $\rho_0(x)$. The correction (128) is non-zero only in the interval $-2 < x < 2$ and diverges at the edges. Moreover, one can notice that the total average spectral density (122) is correctly normalised up to order $\mathcal{O}(1/c)$, given that the integral of the correction (128) over its domain $-2 < x < 2$ is zero.

In Fig. 2, we compare the analytical expression for $\rho_1(x)$ against the data from direct diagonalisation of sparse ER matrices, with bond weights distribution $p_K(K) = \frac{1}{2}\delta_{K,1/\sqrt{c}} + \frac{1}{2}\delta_{K,-1/\sqrt{c}}$, in the case of $c = 50$. We obtained the eigenvalues of 10000 matrices of size $N = 1000$ and discarded the isolated contribution due to the top eigenvalue of such matrices, which lies outside the spectrum. We then organised the remaining $9.99 \times 10^6$ eigenvalues in a normalised histogram and removed from the data the semicircular contribution $\rho_0(x)$ evaluated at the mid-point of each histogram bin. We found a very good agreement between the analytical curve representing $\rho_1(x)$ (red line) and the numerical diagonalisation data (blue dotted line).

As a final remark, we observe that another possible way to extract the large $c$ limit of the average spectral density in the replica formalism would be considering $K = \mathcal{K}/\sqrt{c}$ in Eq. (66),

with the distribution of the rescaled weights being $p_{\mathcal{K}}(\mathcal{K}) = \frac{1}{2}\delta_{\mathcal{K},1} + \frac{1}{2}\delta_{\mathcal{K},-1}$, and expanding the exponential in a Taylor series. This choice would result in the conjugate order parameter being expressed as an expansion in powers of $\frac{1}{c}$, given that the odd powers of $\frac{1}{\sqrt{c}}$ are cancelled by the fact that the odd moments of $p_{\mathcal{K}}(\mathcal{K})$ are zero. Assuming that the order parameter (67) at the saddle point is expressed at the saddle-point as a multivariate factorised zero-mean Gaussian, i.e.

$$\varphi^{\star}(\vec{v}) = \prod_{a=1}^{n} \frac{e^{-\frac{v_a^2}{2\sigma^2}}}{\sqrt{2\pi\sigma^2}}, \tag{129}$$

the leading order of its conjugate results in a quadratic polynomial in the $v_a$, namely

$$i\hat{\varphi}^{\star}(\vec{v}) = -\frac{\langle \mathcal{K}^2 \rangle_{\mathcal{K}}}{2}\sigma^2 \sum_{a=1}^{n} v_a^2, \tag{130}$$

where $\sigma^2$ is determined by the condition $\frac{1}{\sigma^2} = i\lambda_{\varepsilon} + \langle \mathcal{K}^2 \rangle_{\mathcal{K}}\sigma^2$. Using (130) in Eq. (69), one easily obtains the Wigner semicircle. The expansion could also be continued to obtain the corrections in powers of $\mathcal{O}(1/c)$.

# 5 Alternative Replica solution: uncountably infinite superposition of Gaussians

Kühn in [37] suggested a different approach for the average spectral density problem, which completely bypasses (90). At the outset, the treatment in [37] is the same as in [28], but departs from the Bray-Rodgers original derivation at the level of the stationarity conditions (66) and (67). The order parameter $\varphi(\vec{v})$ and its conjugate $i\hat{\varphi}(\vec{v})$ are represented as uncountably infinite superpositions of complex Gaussians, i.e.

$$\varphi(\vec{v}) = \int d\omega\, \pi(\omega) \prod_{a=1}^{n} \frac{e^{-\frac{\omega}{2}v_a^2}}{Z(\omega)}, \tag{131}$$

$$i\hat{\varphi}(\vec{v}) = \hat{c} \int d\hat{\omega}\, \hat{\pi}(\hat{\omega}) \prod_{a=1}^{n} \frac{e^{-\frac{\hat{\omega}}{2}v_a^2}}{Z(\hat{\omega})}, \tag{132}$$

where $Z(x) = \sqrt{\frac{2\pi}{x}}$ and $\pi(\omega)$ and $\hat{\pi}(\hat{\omega})$ are normalised pdfs of the inverse variances $\omega$ and $\hat{\omega}$[5] with $\mathrm{Re}(\omega), \mathrm{Re}(\hat{\omega}) \geq 0$. The constant $\hat{c}$ is chosen to enforce the normalisation of $\hat{\pi}(\hat{\omega})$.

Eq. (131) and (132) are expansions of $\varphi$ and $\hat{\varphi}$ in an over-complete function system. The structure of this ansatz derives from the study of models for amorphous systems. In that context, it was noticed that harmonically coupled systems — such as the model defined by our "Hamiltonian" (5) — admit a solution in terms of superpositions of Gaussians [38, 70]. This ansatz exhibits permutation symmetry among replicas as well as rotational symmetry in replica space, therefore sharing the same symmetries assumed in [28] (see Eq. (72)). The advantage of the ansätze (131) and (132) is that they allow us to extract the leading contribution to the saddle-point of (60) in the limits $N \to \infty$ and $n \to 0$.

The path integral over the $\varphi$ and $\hat{\varphi}$ is thus replaced by a path integral over $\pi$ and $\hat{\pi}$. Therefore, Eq. (60) becomes

$$\langle Z(\lambda)^n \rangle_J \propto \int \mathcal{D}\pi\mathcal{D}\hat{\pi} \exp\left(NS_n[\pi, \hat{\pi}, \lambda]\right), \tag{133}$$

---

[5]We employ the same labels used in the cavity treatment, as the two objects will eventually coincide.

where

$$S_n[\pi, \hat{\pi}, \lambda] = S_1[\pi, \hat{\pi}] + S_2[\pi] + S_3[\hat{\pi}, \lambda], \tag{134}$$

and

$$S_1[\pi, \hat{\pi}] = -\hat{c} - n\hat{c} \int d\pi(\omega) d\hat{\pi}(\hat{\omega}) \log \frac{Z(\omega + \hat{\omega})}{Z(\omega) Z(\hat{\omega})}, \tag{135}$$

$$S_2[\pi] = n \frac{c}{2} \int d\pi(\omega) d\pi(\omega') \left\langle \log \frac{Z_2(\omega, \omega', K)}{Z(\omega) Z(\omega')} \right\rangle_K, \tag{136}$$

$$S_3[\hat{\pi}, \lambda] = \hat{c} + n \sum_{k=0}^{\infty} p_{\hat{c}}(k) \int \{d\hat{\pi}\}_k \log \frac{Z(i\lambda_\varepsilon + \{\hat{\omega}\}_k)}{\prod_{\ell=1}^{k} Z(\hat{\omega}_\ell)}. \tag{137}$$

In the latter expressions, the shorthands $d\pi = d\omega \pi(\omega)$, $\{d\hat{\pi}\}_k = \prod_{\ell=1}^{k} d\hat{\omega}_\ell \hat{\pi}(\hat{\omega}_\ell)$ and $\{\hat{\omega}\}_k = \sum_{\ell=1}^{k} \hat{\omega}_\ell$ have been used. Moreover, $Z_2(\omega, \omega', K) = Z(\omega) Z(\omega' + \frac{K^2}{\omega})$. The function $p_{\hat{c}}(k)$ is the Poisson degree distribution $p_{\hat{c}}(k) = \frac{e^{-\hat{c}} \hat{c}^k}{k!}$, which naturally crops up in the calculation when representing $e^{i\hat{\varphi}^\star(\vec{v})}$ appearing in (67) as a power series. The derivation of (135), (136) and (137) is detailed in Appendix E. We notice that the $\mathcal{O}(1)$ contributions in (135) and (137) cancel each other out, making the action (134) of $\mathcal{O}(n)$.

The stationarity conditions (66) and (67) are replaced by the stationarity conditions w.r.t. $\pi$ and $\hat{\pi}$. They are

$$\frac{\delta S_n}{\delta \pi} = 0 \Rightarrow$$
$$\frac{\hat{c}}{c} \int d\hat{\pi} \log \frac{Z(\omega + \hat{\omega})}{Z(\omega) Z(\hat{\omega})} = \int d\pi' \left\langle \log \frac{Z_2(\omega, \omega', K)}{Z(\omega) Z(\omega')} \right\rangle_K + \frac{\gamma}{c}, \tag{138}$$

and

$$\frac{\delta S_n}{\delta \hat{\pi}} = 0 \Rightarrow$$
$$\int d\pi \log \frac{Z(\omega + \hat{\omega})}{Z(\omega) Z(\hat{\omega})} = \sum_{k=1}^{\infty} p_{\hat{c}}(k) \frac{k}{\hat{c}} \int \{d\hat{\pi}\}_{k-1} \log \frac{Z(i\lambda_\varepsilon + \{\hat{\omega}\}_{k-1} + \hat{\omega})}{Z(\hat{\omega}) \prod_{\ell=1}^{k-1} Z(\hat{\omega}_\ell)} + \frac{\hat{\gamma}}{\hat{c}}, \tag{139}$$

where $\gamma$ and $\hat{\gamma}$ are two Lagrange multipliers to enforce the normalisation condition of $\pi$ and $\hat{\pi}$. The equality (138) is realised by making sure that $\gamma$ satisfies the equality

$$-\frac{\hat{c}}{c} \int d\hat{\pi}(\hat{\omega}) \log Z(\hat{\omega}) = \frac{\gamma}{c}, \tag{140}$$

while the remaining part (which is a function of $\omega$) should satisfy

$$\frac{\hat{c}}{c} \int d\hat{\pi}(\hat{\omega}) \log \frac{Z(\omega + \hat{\omega})}{Z(\omega)} = \int d\pi(\omega') \left\langle \log \frac{Z\left(\omega + \frac{K^2}{\omega'}\right)}{Z(\omega)} \right\rangle_K, \tag{141}$$

where $Z_2(\omega, \omega', K) = Z(\omega') Z(\omega + \frac{K^2}{\omega'})$ has been used. Since Eq. (141) must hold for any value of $\omega$ in order for (138) to be satisfied, one notices that the following definition

$$\hat{\pi}(\hat{\omega}) = \frac{\hat{c}}{c} \int d\pi(\omega) \left\langle \delta\left(\hat{\omega} - \frac{K^2}{\omega}\right) \right\rangle_K, \tag{142}$$

once inserted in the l.h.s. of (142), indeed produces the r.h.s. Likewise, also in (139) a constant part can be isolated,

$$-\int d\pi(\omega)\log Z(\omega) = -\sum_{k=1}^{\infty}p_{\hat{c}}(k)\frac{k}{\hat{c}}\int\{d\hat{\pi}\}_{k-1}\log\prod_{\ell=1}^{k-1}Z(\hat{\omega}_{\ell}) + \frac{\hat{\gamma}}{\hat{c}}\,, \tag{143}$$

and a part that is a function of $\hat{\omega}$, viz.

$$\int d\pi(\omega)\log\frac{Z(\omega+\hat{\omega})}{Z(\hat{\omega})} = \sum_{k=1}^{\infty}p_{\hat{c}}(k)\frac{k}{\hat{c}}\int\{d\hat{\pi}\}_{k-1}\log\frac{Z(i\lambda_{\varepsilon}+\{\hat{\omega}\}_{k-1}+\hat{\omega})}{Z(\hat{\omega})}\,. \tag{144}$$

As before, since (144) must hold for any $\hat{\omega}$, it follows that

$$\pi(\omega) = \sum_{k=1}^{\infty}p_{\hat{c}}(k)\frac{k}{\hat{c}}\int\{d\hat{\pi}\}_{k-1}\delta\left(\omega - (i\lambda_{\varepsilon}+\{\hat{\omega}\}_{k-1})\right)\,. \tag{145}$$

In order for both $\hat{\pi}$ and $\pi$ to be normalised to 1, the condition $\hat{c}=c$ must be imposed.

## 5.1 Average spectral density unfolded

The solutions of the two coupled functional equations

$$\hat{\pi}(\hat{\omega}) = \int d\pi(\omega)\left\langle\delta\left(\hat{\omega} - \frac{K^2}{\omega}\right)\right\rangle_K\,, \tag{146}$$

$$\pi(\omega) = \sum_{k=1}^{\infty}p_c(k)\frac{k}{c}\int\{d\hat{\pi}\}_{k-1}\delta\left(\omega - \left(i\lambda_{\varepsilon}+\sum_{\ell=1}^{k-1}\hat{\omega}_{\ell}\right)\right)\,, \tag{147}$$

represent the saddle-point evaluation of (133). The symbol $p_c(k)$ represents the Poisson degree distribution, which is expected for ER sparse graphs. However, it has been shown in [38,50,53] that the above equations hold *unmodified* also for *any* non-Poissonian degree distributions $p(k)$ within the configuration model framework, as long as the mean degree $\langle k\rangle = c$ is a finite constant, i.e. does not scale with $N$. Unlike (90), the equations (146) and (147) can be very efficiently solved numerically by a population dynamics algorithm (see Section 6). Some remarks are in order.

- Inserting (146) into (147) yields a unique self-consistency equation for $\pi$ that is exactly identical to (34), obtained using the cavity method in the thermodynamic limit. This fact demonstrates once more the equivalence between the replica and cavity methods.

- Alternatively, one could insert (147) into (146), obtaining a single self-consistency equation for $\hat{\pi}$ that reads

$$\hat{\pi}(\hat{\omega}) = \sum_{k=1}^{\infty}p_c(k)\frac{k}{c}\int\{d\hat{\pi}\}_{k-1}\left\langle\delta\left(\hat{\omega} - \frac{K^2}{i\lambda_{\varepsilon}+\sum_{\ell=1}^{k-1}\hat{\omega}_{\ell}}\right)\right\rangle_{\{K\}}\,. \tag{148}$$

The solution of the latter equation via a population dynamics algorithm will be described below in Section 6. While the two approaches are equivalent, here we choose to work with the $\{\hat{\omega}\}$ since the final equation for the spectral density is more naturally expressed in terms of those, as shown in the following.

The pdf $\hat{\pi}(\hat{\omega})$ defined in (148) *fully determines* the average spectral density. Indeed, recalling (2) one gets

$$
\begin{aligned}
\rho(\lambda) &= -\frac{2}{\pi N} \lim_{\varepsilon \to 0^+} \operatorname{Im} \frac{\partial}{\partial \lambda} \langle \log Z(\lambda) \rangle_J \\
&\simeq -\frac{2}{\pi} \lim_{\varepsilon \to 0^+} \operatorname{Im} \lim_{n \to 0} \frac{1}{n} \frac{\partial}{\partial \lambda} S_3[\hat{\pi}, \lambda] \\
&= \frac{1}{\pi} \lim_{\varepsilon \to 0^+} \sum_{k=0}^{\infty} p_c(k) \operatorname{Re} \int \{d\hat{\pi}\}_k \frac{1}{i\lambda_\varepsilon + \{\hat{\omega}\}_k} \\
&= \frac{1}{\pi} \lim_{\varepsilon \to 0^+} \sum_{k=0}^{\infty} p_c(k) \int \{d\hat{\pi}\}_k \frac{\operatorname{Re}[\{\hat{\omega}\}_k] + \varepsilon}{(\operatorname{Re}[\{\hat{\omega}\}_k] + \varepsilon)^2 + (\lambda + \operatorname{Im}[\{\hat{\omega}\}_k])^2},
\end{aligned} \tag{149}
$$

where the latter expression corresponds to Eq. (33) in [37]. We notice that (149) is completely equivalent to (36) if $\hat{\pi}(\hat{\omega})$ is expressed in terms of $\pi(\omega)$ according to (146). All the observations made about (36) hold here as well. The average spectral density as expressed in (149) is evaluated by sampling from a large population distributed according to $\hat{\pi}(\hat{\omega})$: this procedure will also be illustrated in Section 6.

## 5.2 The presence of localised states and the role of $\varepsilon$

The average spectral density (149) can be rewritten in order to isolate singular pure-point contributions from the continuous spectrum. Indeed, defining

$$
P(a, b) = \sum_{k=0}^{\infty} p_c(k) \int \{d\hat{\pi}\}_k \delta(a - \operatorname{Re}[\{\hat{\omega}\}_k]) \delta(b - \operatorname{Im}[\{\hat{\omega}\}_k]), \tag{150}
$$

one finds the identity

$$
\rho(\lambda) = \frac{1}{\pi} \lim_{\varepsilon \to 0^+} \int da\,db\, P(a, b) \frac{a + \varepsilon}{(a + \varepsilon)^2 + (\lambda + b)^2}. \tag{151}
$$

The integrand in (151) becomes singular as $\varepsilon \to 0$ for $a = 0$. These singular contributions can be isolated representing $P(a, b)$ as

$$
P(a, b) = P_0(b)\delta(a) + \tilde{P}(a, b), \tag{152}
$$

yielding for the spectral density

$$
\rho(\lambda) = \frac{1}{\pi} \lim_{\varepsilon \to 0^+} \int db\, P_0(b) \mathcal{L}_\varepsilon(\lambda + b) + \frac{1}{\pi} \lim_{\varepsilon \to 0^+} \int_{a>0} da\,db\, \tilde{P}(a, b) \frac{a + \varepsilon}{(a + \varepsilon)^2 + (\lambda + b)^2}. \tag{153}
$$

Here, $\mathcal{L}_\varepsilon(\lambda + b)$ is a Cauchy distribution with scale (half-width at half-maximum) parameter $\varepsilon$, viz.

$$
\mathcal{L}_\varepsilon(\lambda + b) = \frac{1}{\pi} \frac{\varepsilon}{\varepsilon^2 + (\lambda + b)^2} \xrightarrow[\varepsilon \to 0^+]{} \delta(\lambda + b), \tag{154}
$$

that reduces to a delta-peak in $b = -\lambda$ for any value of $\lambda$ as $\varepsilon \to 0^+$. The spectral density $\rho(\lambda)$ can then be easily evaluated by sampling (see Section 6) from the population of the $a$ and $b$ (i.e. the $\hat{\omega}$). Relying on the law of large numbers and calling $\mathcal{M}$ the number of samples $\{(a_i, b_i)\}$, the two integrals in (153) can indeed be rewritten as

$$
\begin{aligned}
\rho(\lambda) &\simeq \rho_S(\lambda) + \rho_C(\lambda) \\
&\simeq \frac{1}{\mathcal{M}} \sum_{i=0 \wedge a_i=0}^{\mathcal{M}} \mathcal{L}_\varepsilon(\lambda + b_i) + \frac{1}{\pi \mathcal{M}} \sum_{i=0 \wedge a_i>0}^{\mathcal{M}} \frac{a_i + \varepsilon}{(a_i + \varepsilon)^2 + (\lambda + b_i)^2},
\end{aligned} \tag{155}
$$

where $\rho_S(\lambda)$ and $\rho_C(\lambda)$ indicate respectively the singular and the continuous part of the average spectral density. Eq. (155) is equivalent to (162) and corresponds to Eq. (40) in [37].

Figure 3 shows the spectral density obtained for ER matrices with mean degree $c = 2$ and Gaussian weights with zero mean and variance $1/c$. The tails of the distribution and the central peak in $\lambda = 0$ are dominated by localised states, i.e. the eigenvectors corresponding to those values of $\lambda$ have most of their components equal to zero. Given that there is a one-to-one matching between the eigenvectors of graphs and their nodes, a localised state can be also described as an eigenvector that is concentrated on few sites of the graph. Quantitatively, the presence and location of localised states in the spectrum is confirmed by the numerical analysis of the *Inverse Participation Ratio* (IPR) of the eigenvectors in [37]. Given an eigenvector $\boldsymbol{v}$ of a $N \times N$ matrix, its IPR is defined as

$$\text{IPR}(\boldsymbol{v}) = \frac{\sum_{i=1}^{N} v_i^4}{\left(\sum_{i=1}^{N} v_i^2\right)^2} \ . \tag{156}$$

The above definition is independent of the eigenvector's normalisation. The IPR of localised states is $\mathcal{O}(1)$, as opposed to the $\mathcal{O}(N^{-1})$ scaling for delocalised states. Indeed, the numerical study in [36, 37] shows that the $\mathcal{O}(1)$ scaling for IPR is found in correspondence of the tails and around the peak in $\lambda = 0$. Moreover, the IPR analysis makes it possible to relate localised states with the singular contributions to the overall spectrum $\rho(\lambda)$. Indeed, the regions of the spectrum where the IPR is $\mathcal{O}(1)$ are also those where the singular contribution $\rho_S(\lambda)$ dominates over $\rho_C(\lambda)$.

Moreover, when comparing numerical direct diagonalisation and population dynamics results two fundamental aspects must be taken into account:

- On the one hand, the eigenvalues obtained by direct diagonalisation must be suitably binned in order to produce the numerical spectral density profile. The binning procedure smoothens the localised peaks and makes them harder to detect.

- On the other hand, the parameter $\varepsilon$ plays an essential role in highlighting the singular contributions to the spectrum. In the evaluation of (155) only the samples such that $b_i \in [-\lambda - \mathcal{O}(\varepsilon), -\lambda + \mathcal{O}(\varepsilon)]$ for any given value of $\lambda$ contribute to $\rho_S(\lambda)$. Therefore, in order to have enough data for a reliable evaluation of the singular contribution $\rho_S(\lambda)$, one must refrain from using a very small $\varepsilon > 0$ (such as $\varepsilon = \mathcal{O}(10^{-300})$), but rather use a relatively large value $\varepsilon > 0$ (such as $\varepsilon = \mathcal{O}(10^{-3})$), to ensure that $\mathcal{M}\varepsilon\rho_S(\lambda) \gg 1$. Here $\mathcal{M}$ indicates the number of samples used to evaluate the sums in (155).

The effects of the choice of the regulariser $\varepsilon$ are evident in the case $c = 2$ (see Fig. 3), since for such low $c$ localised states prevail in the spectrum. Indeed, this is due to the structure of the graph itself, which is made of a giant cluster component and a collection of isolated finite connected clusters of nodes of any size (see Appendix C). An excellent agreement between direct diagonalisation and population dynamics results is achieved for $\varepsilon = 10^{-3}$ (top left panel). Indeed, in the $\varepsilon = 10^{-3}$ case, the Cauchy peaks related to the singular contributions to the spectral density are broadened into "wider" Cauchy pdfs. On the other hand, when using a smaller regulariser such as $\varepsilon = 10^{-300}$ (top right panel), the spectral density exhibits large fluctuations mainly due to errors that occur when sampling isolated states, as the condition $\mathcal{M}\varepsilon\rho_S(\lambda) \gg 1$ cannot be satisfied. The peaks are superimposed on the continuous curve representing $\rho_C(\lambda)$.

However, the curve $\rho_C(\lambda)$ is unable to capture the tails of the spectral density: this effect is highlighted on a logarithmic scale (bottom panel). This is the typical signature of a *localisation transition*: the tails of the spectral density are dominated by localised states, hence they cannot

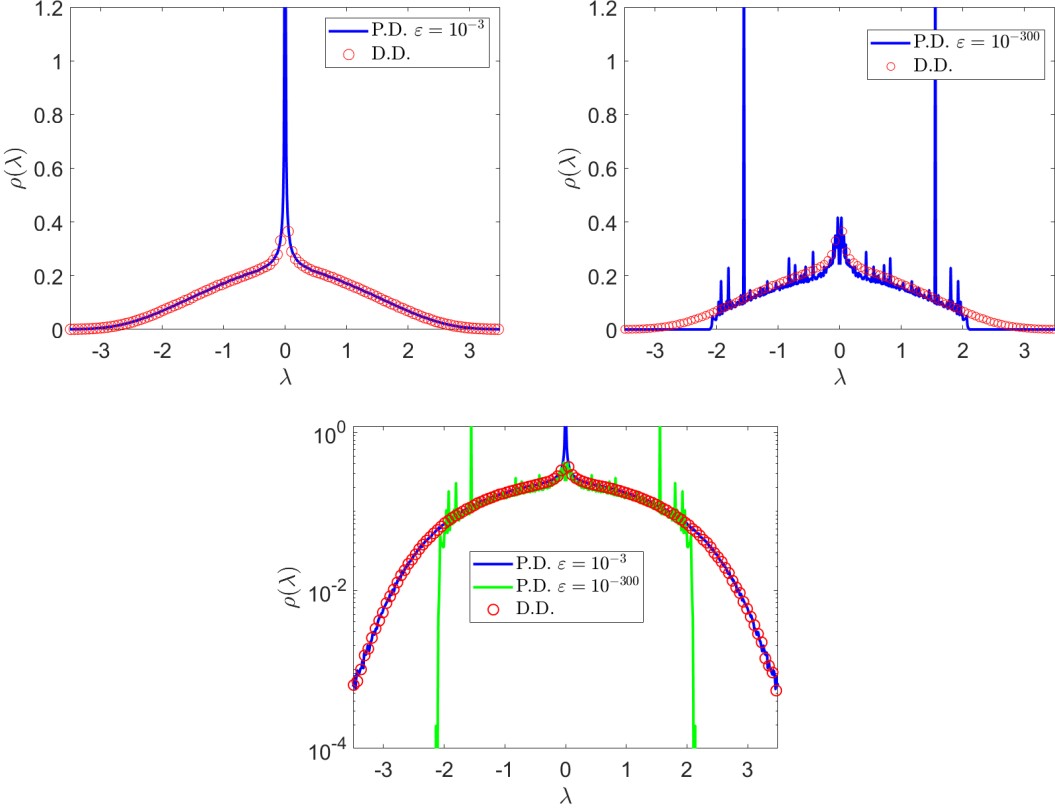

Figure 3: Spectral density of ER matrices with mean degree $c = 2$ and Gaussian bond weights with zero mean and variance $1/c$. In all panels, direct diagonalisation results (red circles) are obtained from a sample of 10000 matrices of size $N = 1000$. **Top left**: population dynamics result obtained with a regulariser $\varepsilon = 10^{-3}$ (solid blue line) vs. the direct diagonalisation results. **Top right**: population dynamics result obtained with a regulariser $\varepsilon = 10^{-300}$ (solid blue line) vs. the direct diagonalisation results. **Bottom**: comparison between population dynamics result obtained with a regulariser $\varepsilon = 10^{-3}$ (solid blue line), population dynamics result obtained with a regulariser $\varepsilon = 10^{-300}$ (solid green line) and direct diagonalisation on a logarithmic scale. An extremely small value of $\varepsilon$ is not able to capture the tails of the spectral density related to localised states, where the singular contributions prevail.

be represented by the continuous part of the spectrum. At the same time, using too small values for $\varepsilon$ makes it impossible to observe the localised state contributions in the tails. If a larger regulariser (hence better local statistics) were employed (top left panel), then the tails could be revealed. For an extensive discussion of these phenomena, see [37].

These effects are less evident in the $c = 4$ case, shown in Fig. 4, since localised states are much less relevant as the mean degree $c$ increases. Indeed it has been shown in [71, 72] that the weight of the delta-peaks related to localised states is an exponentially decreasing function of $c$, hence the peaks tend to disappear and merge into the continuous part of the spectrum as $c$ grows. Moreover, the proportion of isolated nodes and isolated tree-like clusters of nodes in the graph is strongly reduced (see again C). Therefore, in the $c = 4$ case the choice of the regulariser is of lesser importance, and population dynamics simulations run with different values of $\varepsilon$ yield very similar results in the continuous part of the spectrum. This is shown in the left panel of Fig. 4, where population dynamics results obtained with $\varepsilon = 10^{-3}$ (solid blue line) and $\varepsilon = 10^{-300}$ (solid green line) are compared with the numerical diagonalisation of

10000 matrices of size $N = 1000$ (red circles). The peak at $\lambda = 0$ due to isolated nodes can be noticed. The log scale plot reveals that the solid green curve obtained for $\varepsilon = 10^{-300}$ still departs from the blue one obtained for $\varepsilon = 10^{-3}$ at the *mobility edge*, i.e. the value of $\lambda$ at which $\rho_C(\lambda)$ vanishes, hence separating the delocalised from the localised phase. However, the mobility edge for $c = 4$ is located at a larger $\lambda$ than in the $c = 2$ case, in agreement with previous observations [24, 37]. The case of the spectral density of ER matrices with Gaussian couplings with $c = 4$ is also used to show that finite size effects are barely present in the spectral problem, away from the localisation transition. This is confirmed by results in the right panel of Fig. 4 where we compare the numerical diagonalisation of matrices of different size (in particular $N = 1000$ and $N = 4000$) with the same population dynamics simulation run with a large regulariser $\varepsilon = 10^{-3}$, in order to generate sufficient statistics in the tails.

Corroborating the observations of [71, 72], the left panel of Fig. 5 shows the average spectral density for adjacency matrices of ER graphs with Gaussian bond weights with zero mean and variance $1/c$, for growing $c$. The plots are obtained using the population dynamics algorithm with $\varepsilon = 10^{-300}$, in order to highlight the continuous part of the spectrum $\rho_C(\lambda)$. As $c$ grows from 2 to 4, the number of peaks superimposed on the continuous curve is strongly reduced and the location of the mobility edge moves to larger values of $\lambda$, entailing that the relevance of localised states is reduced. As $c$ is further increased, the edge of the continuous spectrum approaches $\lambda = \pm 2$, which are the edges of the Wigner semicircle that would be obtained as $c \to \infty$ with that choice for bond weights statistics.

Finally, as an illustration of the fact that the formalism presented here can be used to obtain the spectral density for other finite mean degree ensembles in the configuration model class, we show in right panel of Fig. 5 the spectral density of the ensemble of adjacency matrices of random regular graphs (RRG), having degree distribution $p(k) = \delta_{k,c}$. We consider the $c = 4$ case. For RRGs adjacency matrices, there are no localised states for any $c > 2$. Conversely, there are mainly localised states for $c = 2$ (see again [37] for a detailed discussion). The population dynamics algorithm perfectly reproduces the Kesten-McKay distribution [45, 46],

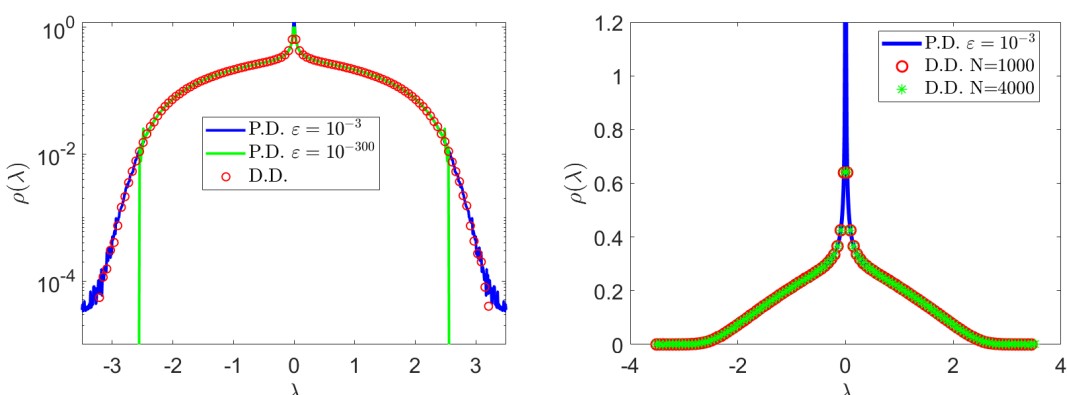

Figure 4: Spectral density of ER matrices with mean degree $c = 4$ and Gaussian bond weights with zero mean and variance $1/c$. **Left**: population dynamics results obtained with a regulariser $\varepsilon = 10^{-3}$ (solid blue line) and $\varepsilon = 10^{-300}$ (solid green line) vs. direct diagonalisation results (red circles) obtained from a sample of 10000 matrices of size $N = 1000$. The plot is on a logarithmic scale. **Right**: comparison between population dynamics result with $\varepsilon = 10^{-3}$ (solid blue line), direct diagonalisation results obtained from a sample of 10000 matrices of size $N = 1000$ (red circles) and direct diagonalisation results obtained from a sample of 2500 matrices of size $N = 4000$ (green stars).

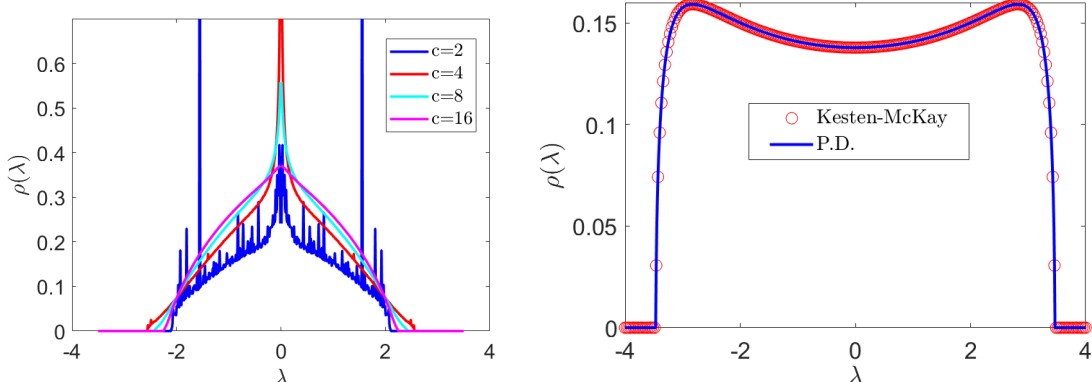

Figure 5: **Left**: spectral density of adjacency matrices of ER graphs with Gaussian bond weights with zero mean and variance $1/c$, for growing values of $c$, obtained with population dynamics algorithm using a regulariser $\varepsilon = 10^{-300}$ to highlight the continuous part of the spectrum. **Right**: spectral density of adjacency matrices of random regular graphs with coordination $c = 4$. The population dynamics result (solid blue line) is compared to the analytical expression of the spectral density, found in [45, 46], known as Kesten-McKay pdf (red circles).

given by the analytical formula

$$\rho(\lambda) = \frac{c\sqrt{4(c-1)-\lambda^2}}{2\pi(c^2-\lambda^2)} \quad \text{for } |\lambda| \leq 2\sqrt{(c-1)} \,. \tag{157}$$

We remark that (157) can be derived analytically within the formalism of Section 5.1 employing a "peaked" ansatz for the distribution of inverse variances as $\hat{\pi}(\hat{\omega}) = \delta(\hat{\omega} - \bar{\omega})$. This is shown in Appendix F.

## 6 Population dynamics algorithm

In this section, we sketch the stochastic population dynamics algorithm that allows us to solve the self-consistency equation (148) and the sampling procedure to evaluate (149). This kind of algorithm is widely used in the study of spin glasses [73, 74]. This procedure is general and allows to solve every equation having the same structure as (148) (including for instance (34)).

In order to solve (148), one represents the pdf $\hat{\pi}(\hat{\omega})$ in terms of a population of $N_P$ complex values $\{(\hat{\omega}_i)\}_{1 \leq i \leq N_P}$, which are assumed to be sampled from that pdf. Given that the true pdf is initially unknown, a starting population is randomly initialised with $\text{Re}[\hat{\omega}_i] > 0$. Then, a stochastic algorithm for which the solution of Eq. (148) is the unique stationary solution is constructed.

To start, we fix $\varepsilon = 10^{-300}$. Indeed, when solving (148), we may choose $\varepsilon$ to be as small as possible in order not to bias the values of the $\hat{\omega}$ [6]. Moreover, we define a set $I$ of equally spaced real positive numbers, starting at zero. The parameter $\lambda$ will take values in $I$. The distance between two consecutive values in $I$, denoted by $\Delta\lambda$, represents the $\lambda$-scan. For the plots shown in this paper, we have employed $\Delta\lambda = \mathcal{O}(10^{-3})$. However, the mesh precision can be tuned depending on the desired resolution of the spectrum. Since the average spectral density as expressed by Eq. (151) (or equivalently Eq. (155)) is an even function of $\lambda$, it

---

[6]This morally corresponds to considering the $\varepsilon \to 0^+$ limit in Eq. (148).

can be evaluated in the interval $I$ and then simply mirrored w.r.t. 0 to obtain the full spectral density shape. As a last remark, it is convenient to normalise the bond weights drawn from $p_K(K)$ by $\sqrt{c}$, where $c$ is the mean degree.

Given these initial remarks, the stochastic algorithm consists in iterating the following steps until a statistically stationary population is obtained, for any given $\lambda \in I$.

1. Generate a random $k$ according to the distribution $\frac{k}{c}p(k)$, where $p(k)$ is the degree distribution of interest and $c = \langle k \rangle$.

2. Generate $K$ from the bond weights pdf $p_K(K)$.

3. Select $k-1$ elements $\hat{\omega}_\ell$ from the population at random, then compute

$$\hat{\omega}^{(new)} = \frac{K^2}{i\lambda_\varepsilon + \sum_{\ell=1}^{k-1} \hat{\omega}_\ell} \, , \tag{158}$$

   which is the equality enforced by the delta function in (148) . Replace a randomly selected population member $\hat{\omega}_j$ (where $j \in \{1, \dots, N_p\}$) with $\hat{\omega}^{(new)}$.

4. Return to (1.).

A *sweep* is completed when every member of the population $\hat{\omega}_j$ with $j = 1, \dots, N_P$ has been updated once according to the previous steps. We denote the $i$-th sweep for a given $\lambda$ as $S_i(\lambda)$. A sufficient number $N_{\text{eq}}$ of sweeps is needed to equilibrate the population. Stationarity can be assessed by looking at the sample estimate of the first moments of the $\hat{\omega}$ variables.

The population dynamics algorithm can also be employed for the sampling procedure that allows one to numerically evaluate (149) (and in a similar fashion (36)). Once (for a given value of $\lambda$) the population $\{(\hat{\omega}_i)\}_{1 \le i \le N_p}$ has been brought to convergence after $N_{\text{eq}}$ equilibration sweeps, a number $N_{\text{meas}}$ of so-called *measurement sweeps $M(\lambda)$* is performed. Here, $N_{\text{meas}}$ is the number of the measurement sweeps.

Each measurement sweep $M_j(\lambda)$ ($j = 1, \dots, N_{\text{meas}}$) can be divided into two parts. In the first part, the population in equilibrium is updated via a sweep $S_j(\lambda)$, as described before. The second part is the actual measurement part $m_j(\lambda)$, involving the following steps. Two (real) empty arrays $\{a_i\}_{\{1 \le i \le N_{\text{sam}}\}}$ and $\{b_i\}_{\{1 \le i \le N_{\text{sam}}\}}$ of size $N_{\text{sam}}$ are initialised. Here, $N_{\text{sam}}$ is the number of samples per measurement sweep. Each of the $a_i$ and the $b_i$ will eventually play the role of the real and the imaginary parts of sums of the $\hat{\omega}_\ell$, respectively. Then, for $i = 1, \dots, N_{\text{sam}}$:

1. Generate a random $k$ according to the degree distribution $p_c(k)$ (or in general $p(k)$)

2. Select $k$ elements $\hat{\omega}_\ell$ from the population $\{\omega_i\}_{1 \le i \le N_p}$ at random and compute

$$x_i = \sum_{\ell=1}^{k} \hat{\omega}_\ell \, . \tag{159}$$

3. Compute

$$a_i = \text{Re}[x_i], \tag{160}$$
$$b_i = \text{Im}[x_i] \, . \tag{161}$$

When all $N_{\text{meas}}$ measurement sweeps $M_j(\lambda)$ have been completed (typically, $N_{\text{meas}} \approx 10^4$), the resulting $N_{\text{meas}}$ arrays of the type $\{a_i\}_{\{1 \le i \le N_{\text{sam}}\}}$ (respectively $\{b_i\}_{\{1 \le i \le N_{\text{sam}}\}}$) are merged together, yielding a unique large array $\{A_j\}_{\{1 \le j \le \mathcal{M}\}}$ (respectively $\{B_j\}_{\{1 \le j \le \mathcal{M}\}}$) of size $\mathcal{M} = N_{\text{sam}}$

$\times N_{\mathrm{meas}}$, which is the total number of samples (typically, we choose $N_{\mathrm{sam}}$ such that $\mathcal{M}$ is $\mathcal{O}(10^7)$ for the chosen value of $N_{\mathrm{meas}}$). Therefore we can eventually compute

$$\rho(\lambda) = \frac{1}{\pi \mathcal{M}} \sum_{j=1}^{\mathcal{M}} \frac{A_j + \varepsilon}{(A_j + \varepsilon)^2 + (B_j + \lambda)^2} \, , \tag{162}$$

which represents the contribution to the average spectral density for a given value of $\lambda$. It should be noticed that the sample size $\mathcal{M}$ that one employs to compute the sum (162) is completely unrelated to the population size $N_P$. The evaluation of the sample average (162) requires a careful choice of $\varepsilon$, since the value of $\varepsilon$ in (162) will determine the width of the Cauchy distributions approximating the delta-peaks in the spectrum (see the discussion in Section 5.2). In general, the value of $\varepsilon$ employed in (162) can be larger (up to $\varepsilon = \mathcal{O}(10^{-3})$) than the value $\varepsilon = 10^{-300}$ chosen for the equilibration sweeps in (158), in order to have sufficient statistics to faithfully represent the singular part of the spectrum. Eq. (162), corresponding to Eq. (40) in [37], is a discretised version of (151). This step concludes the sampling algorithm for a given value of $\lambda$. The full sampling algorithm can be summarised as follows.

---

**Algorithm 1** Population dynamics sampling algorithm

---

1: **for** $\lambda \in I$ **do**
2:      **for** $e = 1, ..., N_{\mathrm{eq}}$ **do**
3:          $S_e(\lambda)$                                        ▷ Equilibration sweeps.
4:      **end for**
5:      **for** $j = 1, ..., N_{\mathrm{meas}}$ **do**
6:          $S_j(\lambda)$
7:          $m_j(\lambda)$          ▷ $S_j(\lambda)$ and $m_j(\lambda)$ jointly form the measurement sweep $M_j(\lambda)$.
8:      **end for**
9:      Compute (162) for given $\lambda$
10: **end for** .

---

An example of population dynamics algorithm for the spectra of ER matrices is available upon request.

# 7 Conclusions

In summary, we have provided a pedagogical and comprehensive overview of the computation of the average spectral density of sparse symmetric random matrices. We started with the celebrated Edwards-Jones formula (2) and outlined its proof. The formula allows to recast the determination of the density of states of a $N \times N$ matrix into the calculation of the average free energy of a system of $N$ interacting particles at equilibrium, described by a Gibbs-Boltzmann distribution at imaginary inverse temperature. Therefore, techniques from the statistical physics of disordered systems, such as the replica method, can be employed to correctly deal with the calculation of the average free energy (see Eq. (16)) that features in the Edwards-Jones formula. The replica method was indeed the strategy used in the seminal work of Bray and Rodgers, which represents the first attempt to obtain the spectral density for matrices with ER connectivity. We have reproduced their calculations in detail, showing how to derive the integral equation (90) whose solution still represents a challenging open problem. We also described how to obtain the Wigner semicircle as the leading order of the large mean degree expansion of Eq. (90), as well as the first $1/c$ correction.

Considering sparse tree-like graphs within the Edwards-Jones framework, we have described how to apply the cavity method to the spectral problem for single instances. The

cavity method circumvents the averaging of the free energy by making the associated Gibbs-Boltzmann distribution the target of its analysis. We have demonstrated that in this context the only ingredients needed to compute the spectral density are the inverse variances of each of the $N$ marginal pdfs of the Gibbs-Boltzmann distribution (see Eq. (22)). These inverse variances are easily obtained in terms of a set of self-consistency equations (29) for the cavity inverse variances. Moreover, we have explained how in the thermodynamic limit the cavity single-instance recursions give rise to a self-consistency integral equation (34) for the pdf of the inverse cavity variances, in terms of which the average spectral density (36) is fully determined at the ensemble level.

We have also illustrated an alternative replica derivation, where the high-temperature replica symmetry ansatz employed by Bray and Rodgers is realised by assuming that the order parameter and its conjugate are expressed through an infinite superposition of zero-mean complex Gaussians, with random inverse variances (see Eq. (131) and (132)). We showed that the two coupled integral equations (146) and (147) that define the pdfs of the aforementioned inverse variances reduce to a unique self-consistency equation (148), which is completely equivalent to Eq. (34) found within the cavity treatment in the thermodynamic limit. In the replica framework, too, the average spectral density (149) depends only on this single pdf defined in (148). Therefore, once again the equivalence between the cavity and replica approaches is confirmed. Indeed, both methods permit to express the average spectral density as a weighted sum of local contributions coming from nodes of different degree $k$. On the practical side, the average spectral density is obtained by sampling from a large population of complex numbers distributed according to the pdf of the inverse variances (148) (or equivalently (34)). We remark that both approaches are not restricted to ER graphs, but can handle any degree distribution with finite mean degree.

The essential tool for solving self-consistency equations of the kind of Eq. (148) and performing the sampling procedure to evaluate the spectral density (149) is a stochastic population dynamics algorithm. We give a detailed description of the algorithm along with pratical tips to implement it. Results obtained with the population dynamics algorithm are in excellent agreement with the numerical diagonalisation of large weighted adjacency matrices of tree-like graphs, provided that the correct choice of the value of the regulariser $\varepsilon$ used in the algorithm is made. Indeed, we thoroughly describe the important role of $\varepsilon$ in unveiling the contribution of the localised states to the spectral density, also in connection with the mean degree and structure of the graph. We are also able to show that the spectral density does not significantly suffer from finite size effects, away from any localisation transition.

In line with the pedagogical gist of this work, we include a number of appendices where we provide background information on graphs, discussion of technical aspects, and quick proofs of the identities that have been used in the main text.

We believe that there are still open pathways for further research in this field. One might be the investigation of the properties of the eigenvectors of sparse random matrices through the analysis of the statistics of the local resolvent using the cavity method. As mentioned in Section 3.5, there is a connection between the resolvent of a matrix $J$ (43), its eigenvectors, and the marginal inverse variances $\omega_i$ defined in Eq. (31). Indeed, given a $N \times N$ matrix $J$ with eigenpairs $\{(\lambda_\alpha, \boldsymbol{u}_\alpha)\}_{\alpha=1,\ldots,N}$, whose resolvent is $G(z) = (z\mathbb{1} - J)^{-1}$, from Eq. (46), the following identity holds

$$\varepsilon \, \mathrm{Im}\, G(\lambda - \mathrm{i}\varepsilon)_{ii} = \varepsilon \pi \sum_{\alpha=1}^{N} \delta_\varepsilon(\lambda - \lambda_\alpha) u_{i\alpha}^2 \,, \tag{163}$$

for $\lambda \in \mathbb{R}$ and for any $i = 1, \ldots, N$. The function $\delta_\varepsilon(x - x_0) = \frac{1}{\pi} \frac{\varepsilon}{(x - x_0)^2 + \varepsilon^2}$ approximates the Dirac delta as $\varepsilon \to 0$. When $\lambda = \lambda_\alpha$, then $\varepsilon \, \mathrm{Im}\, G(\lambda - \mathrm{i}\varepsilon)_{ii} \simeq u_{i\alpha}^2 + \mathcal{O}(\varepsilon^2)$. On the other

hand, considering Eq. (46) for just a single instance (i.e. omitting the ensemble average) and comparing it with Eq. (32), one eventually observes that

$$\text{Re}\left[\frac{1}{\omega_i}\right] = \pi \sum_{\alpha=1}^{N} \delta_\epsilon(\lambda - \lambda_\alpha) u_{i\alpha}^2, \tag{164}$$

where the $\omega_i$ are defined in Eq. (31). Eq. (164) suggests that the distribution of the squares of the eigenvectors can be obtained in terms of that of the $\omega_i$. While this suggestion appears to be warranted for eigenvectors corresponding to isolated eigenvalues, preliminary results indicate that it may be less reliable for the continuous component of the spectrum, which exists above the percolation threshold. It is likely that a combination of two facts is responsible for this observation. First, cycles will appear in random graph ensembles above the percolation threshold, as a consequence of which the decorrelation assumptions used in the cavity method will be only approximate and no longer exact. For a wide variety of problems in fields as diverse as the physics of spin-glasses, optimisation or percolation, this aspect is well known not to cause any problems in applications of the cavity method. This is generally attributed to the fact that the tree-like approximation becomes asymptotically exact in the thermodynamic limit for networks in the configuration model class that are on average finitely connected. However, the problem of eigenvector components may be different in this respect, and this is where the second fact comes into play. Due to the occurrence of quasi-degeneracies in the continuous spectrum, it is conceivable that hybridisations of near degenerate eigenstates rather than the true eigenstates may appear in any approximate evaluation of Eq. (164). Indeed, if the $u_{i\alpha}$ appearing in Eq. (164) were in an approximate evaluation in fact replaced by linear combinations of components of eigenvectors corresponding to *several* nearly degenerate eigenvalues, then individual components would be modified, whereas *sums of squares of components* would remain unaffected due to orthonormality of the underlying true eigenvectors. This could naturally reconcile the seemingly contradictory observations that eigenvalue densities can be evaluated with great precision using the cavity method, whereas this may not be true for individual eigenvector components.

Another topic of great importance would be the understanding of finite-size effects in the immediate vicinity of localisation transitions, which are expected to affect results as in other continuous phase transitions. The study of these phenomena for spectra of sparse random matrices also provides a framework for the investigation of Anderson localisation, which has recently regained new interest in view of its connection to systems exhibiting many-body localisation (see [75] for a recent review). Despite a variety of works concerning the localisation transition in spectra of sparse symmetric random matrices [15, 23, 24] and studies on the localisation of their top eigenvector [76, 77], a systematic approach for the analysis of the finite size effects at the transition is still needed. Steps in this direction have been done in [78, 79], where the authors focus on finite size effects concerning Anderson localisation on RRGs.

In connection with the localisation transition, another unexplored field that deserves further investigation is the behaviour of the population dynamics algorithm at the transition. Indeed, as pointed out in [37] it exhibits critical slowing down and long autocorrelation time, which in turn affect the quality of the averages. Moreover, as highlighted in [80] for glassy systems on networks, the population dynamics results at the transition are also affected by finite population size effects. Therefore a systematic analysis of how the algorithm is influenced by the crossover between the delocalised and the localised phase would be extremely insightful not only for the spectral problem per se but in general whenever this kind of algorithm is employed.

## Acknowledgements

PV is grateful to Fabian Aguirre Lopez and Ton Coolen for many useful discussions. The authors acknowledge Rémi Monasson and Enzo Nicosia for insightful suggestions.

**Funding information** The authors acknowledge funding by the Engineering and Physical Sciences Research Council (EPSRC) through the Centre for Doctoral Training in Cross Disciplinary Approaches to Non-Equilibrium Systems (CANES, Grant Nr. EP/L015854/1).

## A Sokhotski-Plemelj formula

The Sokhotski-Plemelj identity is

$$\lim_{\varepsilon \to 0^+} \frac{1}{x \pm i\varepsilon} = \Pr\left(\frac{1}{x}\right) \mp i\pi\delta(x). \tag{165}$$

It is employed for the solution of some improper integrals. A quick proof for a real test function $g(x)$ follows. We have

$$\lim_{\varepsilon \to 0^+} \int_{-\infty}^{\infty} dx\, \frac{g(x)}{x \pm i\varepsilon} = \lim_{\varepsilon \to 0^+} \int_{-\infty}^{\infty} dx\, x \frac{g(x)}{x^2 + \varepsilon^2} \mp i\pi \lim_{\varepsilon \to 0^+} \int_{-\infty}^{\infty} dx\, g(x) \frac{1}{\pi} \frac{\varepsilon}{x^2 + \varepsilon^2}, \tag{166}$$

where the real and imaginary part of the integrand have been separated. The first integral on the r.h.s. of (166) can be written as a Cauchy principal value, viz.

$$\lim_{\varepsilon \to 0^+} \int_{-\infty}^{\infty} dx\, x \frac{g(x)}{x^2 + \varepsilon^2} = \lim_{\varepsilon \to 0^+} \int_{-\infty}^{\infty} dx\, \frac{g(x)}{x} \frac{x^2}{x^2 + \varepsilon^2} \tag{167}$$

$$= \lim_{\varepsilon \to 0^+} \left( \int_{-\infty}^{-\varepsilon} dx\, \frac{g(x)}{x} + \int_{\varepsilon}^{\infty} dx\, \frac{g(x)}{x} \right) = \left[ \Pr\left(\frac{1}{x}\right) \right](g). \tag{168}$$

The second integral on the r.h.s of (166) reduces to

$$\lim_{\varepsilon \to 0^+} \int_{-\infty}^{\infty} dx\, g(x) \frac{1}{\pi} \frac{\varepsilon}{x^2 + \varepsilon^2} = \int_{-\infty}^{\infty} dx\, g(x)\delta(x) = g(0), \tag{169}$$

where $\lim_{\varepsilon \to 0^+} \frac{1}{\pi} \frac{\varepsilon}{x^2+\varepsilon^2} = \delta(x)$ has been employed.

## B The principal branch of the complex logarithm

The logarithm in the complex plane is in general a multi-valued function. Whenever a well defined, single-valued function is needed, the *principal branch* of the complex logarithm can be considered. It is denoted by "Log" and defined such that for any $z \in \mathbb{C}$ with $r = |z|$,

$$\text{Log}(z) = \ln(r) + i\text{Arg}(z) \quad \text{with Arg}(z) \in ]-\pi, \pi]. \tag{170}$$

The function $\text{Arg}(z)$ denotes the *principal value* of the argument of the complex number $z$. In particular, given $z = re^{i\theta} \in \mathbb{C}$, the argument of $z$ is given by $\arg(z) = \theta$ and is in general a multi-valued function. The single-valued principal argument $\text{Arg}(z)$ is related to $\arg(z)$ via the following relation,

$$\text{Arg}(z) = \arg(z) + 2\pi \left\lfloor \frac{1}{2} - \frac{\arg(z)}{2\pi} \right\rfloor, \tag{171}$$

where the symbol $\lfloor ... \rfloor$ denotes the floor operation, i.e. $\lfloor x \rfloor$ is the integer number such that $x - 1 < \lfloor x \rfloor \le x$ for $x \in \mathbb{R}$.

In general $\mathrm{Log}\, e^z \ne z$ for $z \in \mathbb{C}$. Indeed, for any $z = x + iy \in \mathbb{C}$ the following property holds:

$$
\begin{aligned}
\mathrm{Log}(e^z) &= \mathrm{Log}|e^z| + i\mathrm{Arg}(e^z) = \mathrm{Log}(e^x) + i\mathrm{Arg}(e^{iy}) \\
&= x + i\left\{\arg(e^{iy}) + 2\pi \left\lfloor \frac{1}{2} - \frac{\arg(e^{iy})}{2\pi} \right\rfloor\right\} \\
&= x + iy + 2\pi i \left\lfloor \frac{1}{2} - \frac{y}{2\pi} \right\rfloor \\
&= z + 2\pi i \left\lfloor \frac{1}{2} - \frac{\mathrm{Im}[z]}{2\pi} \right\rfloor ,
\end{aligned}
\tag{172}
$$

where $\mathrm{Log}(x) = \ln(x)$ for $x \in \mathbb{R}$ and the definition (171) has been used for the principal value of the argument $\mathrm{Arg}(z)$.

## C  Erdős-Rényi graphs

The Erdős-Rényi (ER) graph is the prototypical example of a random graph, introduced by Erdős and Rényi in [81,82]. It is the simplest and most studied uncorrelated undirected random network. It can be denoted by $G(N, p)$, where $N$ is the number of nodes and $p \in [0, 1]$ is the probability that any two nodes (there are $N(N-1)/2$ possible pairs, hence possible links) are connected. In other words, $p$ is the probability that a link exists *independently* from the others. In formulae, the probability that a link exists between nodes $i$ and $j$ is

$$
p_C(c_{ij}) = p\,\delta_{c_{ij},1} + (1 - p)\,\delta_{c_{ij},0} .
\tag{173}
$$

All properties of the ER model depend on the two parameters $N$ and $p$. Its degree distribution is binomial, viz.

$$
\Pr[\text{a random node has degree } k] = p(k) = \binom{N-1}{k} p^k (1-p)^{N-1-k} .
\tag{174}
$$

Indeed, a node has degree $k$ if it is connected to $k$ nodes (the probability of this event being $p^k$) and at the same it is not connected to all the remaining $N-1-k$ nodes (the probability of this event being $(1-p)^{N-1-k}$). The binomial coefficient accounts for the fact that the specific subset of $k$ nodes we choose out of the remaining $N-1$ does not matter. The mean degree is then $c = p(N-1)$. In the limit $N \to \infty$ where $N - 1 \simeq N$ and keeping $c = Np$ constant, the binomial distribution in (174) converges to the Poisson distribution,

$$
p_c(k) = \frac{c^k e^{-c}}{k!} .
\tag{175}
$$

The condition for this limit to hold is exactly verified in the sparse ER ensemble that we consider in our analysis. Indeed, we explicitly ask that the mean degree $c$ be a finite constant, hence ensuring that $p = \frac{c}{N} \to 0$ as $N \to \infty$. The Poisson distribution in (175) is decaying exponentially for large degree $k$.

The structure of an ER graph and in particular the existence of a giant component depend on the value of $p$ [82]. The giant component of a graph is the largest connected component (i.e. cluster of nodes) in the graph, containing a finite fraction of the total $N$ nodes. In a connected component, every two nodes are connected by a path, whereas there are no connections between two nodes belonging to two different components. We have the following properties [83,84]:

- For $p < \frac{1}{N}$ (i.e. $c < 1$), the probability of having a giant component is zero. Indeed, almost surely there are no connected components with size larger than $\mathcal{O}(\ln(N))$. The graph can be described as a disjoint union of trees and unicycle components, i.e. trees with an extra link forming a cycle.

- For $p > \frac{1}{N}$ (i.e. $c > 1$), the probability of having a giant component is 1. Almost surely, the graph will have a unique giant component whose size is $\mathcal{O}(N)$ and contains cycles of any length, while the remaining smaller components (typically trees and unicycles) have at most size $\mathcal{O}(\ln(N))$.

- $p = \frac{1}{N} = p_c$ represents the *percolation threshold* as it separates the two regimes: indeed at $p = p_c$ (i.e. $c = 1$) most of the isolated components for $c < 1$ merge together, giving rise to a giant component of size $\mathcal{O}(N^{2/3})$. As the (constant) mean degree $c > 1$ increases, the smaller components join the giant component, which then becomes $\mathcal{O}(N)$ in size. The smaller the size of the isolated components, the longer they will survive the merging process.

- For $p \leq \frac{\ln(N)}{N}$, the graph contains isolated nodes almost surely, hence it is disconnected. As soon as $p > \frac{\ln(N)}{N}$, the graph becomes connected, as the isolated nodes attach to the giant component entailing that every pair of nodes in the graph is connected by a path. The value $p = \frac{\ln(N)}{N}$ is then a threshold for the *connectivity* of the graph.

The structural properties of the graph are reflected in the spectrum. Indeed, the variety of peaks in the spectrum related to singular contributions are due to isolated nodes and isolated finite clusters of nodes that are still present for finite constant $c > 1$, alongside with the giant component.

The ER graph can also be seen as a model of link percolation [85]. Indeed, ER graphs can be generated also starting from a fully connected graph and removing links at random with constant probability $1 - p$.

An algorithm for the generation of the adjacency matrix of any generic random graphs within the configuration model is described in Section 8.1 and detailed in Appendix J.5 (Algorithm 27) of [86]. A simple code for the generation of single instances of adjacency matrices of ER graphs is available upon request.

## D  How to perform the average (54)

The goal is to perform the average

$$\left\langle \exp\left( \frac{\mathrm{i}}{2} \sum_{i,j=1}^{N} \sum_{a=1}^{n} v_{ia} J_{ij} v_{ja} \right) \right\rangle_J \tag{176}$$

w.r.t. the joint distribution of the matrix entries

$$P(\{J_{ij}\}) = \prod_{i<j} p_C(c_{ij}) \delta_{c_{ij}, c_{ji}} \prod_{i<j} p_K(K_{ij}) \delta_{K_{ij}, K_{ji}}, \tag{177}$$

where

$$p_C(c_{ij}) = \frac{c}{N} \delta_{c_{ij}, 1} + \left( 1 - \frac{c}{N} \right) \delta_{c_{ij}, 0} \tag{178}$$

represents the ER connectivity distribution, and $p_K(K_{ij})$ is the bond weight pdf. The average is computed for large $N$ as follows,

$$
\begin{aligned}
\left\langle \exp\left( \frac{i}{2} \sum_{i,j=1}^N \sum_{a=1}^n v_{ia} J_{ij} v_{ja} \right) \right\rangle_J &= \left\langle \prod_{i<j} \exp\left( i \sum_{a=1}^n v_{ia} c_{ij} K_{ij} v_{ja} \right) \right\rangle_{\{c\},\{K\}} \\
&= \prod_{i<j} \left\langle \exp\left( i \sum_{a=1}^n v_{ia} c_{ij} K_{ij} v_{ja} \right) \right\rangle_{c,K} \\
&= \prod_{i<j} \left[ 1 + \frac{c}{N} \left( \langle e^{iK \sum_a v_{ia} v_{ja}} \rangle_K - 1 \right) \right] \\
&\simeq \exp\left[ \frac{c}{2N} \sum_{i,j=1}^N \left( \left\langle e^{iK \sum_a v_{ia} v_{ja}} \right\rangle_K - 1 \right) \right] ,
\end{aligned}
\tag{179}
$$

where the subscripts $\{c\}$ and $\{K\}$ respectively denote averaging w.r.t. the joint pdfs of the $\{c_{ij}\}$ and the bond weights $\{K_{ij}\}$, whereas the non-bracketed subscripts $c$ and $K$ refers to the average over a single random variable drawn from $p_C(c)$ and $p_K(K)$, respectively. Moreover, in the second line we have used independence of the random variables and in the last line we have re-exponentiated the product and the factor $1/2$ prevents from over-counting symmetric terms in the double sum.

# E   The action $S_n$ in terms of $\pi$ and $\hat{\pi}$

The following action is derived in Section 5,

$$
S_n[\pi, \hat{\pi}, \lambda] = S_1[\pi, \hat{\pi}] + S_2[\pi] + S_3[\hat{\pi}, \lambda] .
\tag{180}
$$

The contributions (135), (136) and (137) are obtained from (62), (63) and (64) respectively, using the saddle-point expressions (131) and (132) for the order parameter $\varphi^\star(\vec{v})$ and its conjugate $i\hat{\varphi}^\star(\vec{v})$. Defining the shorthands $d\pi(\omega) = d\omega\,\pi(\omega)$, $\{d\hat{\pi}\}_k = \prod_{\ell=1}^k d\hat{\omega}_\ell \hat{\pi}(\hat{\omega}_\ell)$, $\{\hat{\omega}\}_k = \sum_{\ell=1}^k \hat{\omega}_\ell$ and $Z(x) = \int dv\, e^{-\frac{x}{2} v^2} = \sqrt{\frac{2\pi}{x}}$, one finds

$$
\begin{aligned}
S_1[\pi, \hat{\pi}] &= -\hat{c} \int d\pi(\omega) d\hat{\pi}(\hat{\omega}) \int d\vec{v} \prod_{a=1}^n \frac{e^{-\left(\frac{\omega+\hat{\omega}}{2}\right) v_a^2}}{Z(\omega) Z(\hat{\omega})} \\
&= -\hat{c} \int d\pi(\omega) d\hat{\pi}(\hat{\omega}) \left( \frac{Z(\omega+\hat{\omega})}{Z(\omega) Z(\hat{\omega})} \right)^n \\
&\simeq -\hat{c} - n\hat{c} \int d\pi(\omega) d\hat{\pi}(\hat{\omega}) \log\left( \frac{Z(\omega+\hat{\omega})}{Z(\omega) Z(\hat{\omega})} \right) ,
\end{aligned}
\tag{181}
$$

where we have used a small $n$ expansion in the last line. Concerning $S_2$, one has

$$
\begin{aligned}
S_2[\pi] &= \frac{c}{2} \int d\vec{v} d\vec{v}' \int d\pi(\omega) d\pi(\omega') \prod_{a=1}^n \frac{e^{-\frac{\omega}{2} v_a^2}}{Z(\omega)} \frac{e^{-\frac{\omega'}{2} v_a'^2}}{Z(\omega')} \left( \langle e^{iK \sum_a v_a v_a'} \rangle_K - 1 \right) \\
&= \frac{c}{2} \int d\pi(\omega) d\pi(\omega') \left[ \left\langle \left( \frac{Z_2(\omega, \omega', K)}{Z(\omega) Z(\omega')} \right)^n \right\rangle_K - 1 \right] \\
&\simeq n\frac{c}{2} \int d\pi(\omega) d\pi(\omega') \left\langle \log \frac{Z_2(\omega, \omega', K)}{Z(\omega) Z(\omega')} \right\rangle_K ,
\end{aligned}
\tag{182}
$$

where we have used $Z_2(\omega, \omega', K) = \int d\nu d\nu' e^{-\frac{\omega}{2}\nu^2 - \frac{\omega'}{2}\nu'^2 + iK\nu\nu'}$ and again a small $n$ expansion. Concerning $S_3$, one gets

$$
\begin{aligned}
S_3[\hat{\pi}, \lambda] &= \text{Log} \int d\vec{\nu}\, e^{-i\frac{\lambda}{2}\sum_a \nu_a^2 + i\hat{\varphi}(\vec{\nu})} \\
&= \text{Log} \int d\vec{\nu}\, e^{-i\frac{\lambda}{2}\sum_a \nu_a^2} \sum_{k=0}^{\infty} \frac{(i\hat{\varphi}(\vec{\nu}))^k}{k!} \\
&= \text{Log} \sum_{k=0}^{\infty} \frac{\hat{c}^k}{k!} \int d\vec{\nu}\, e^{-i\frac{\lambda}{2}\sum_a \nu_a^2} \int \{d\hat{\pi}\}_k \prod_{\ell=1}^{k}\prod_{a=1}^{n} \frac{e^{-\frac{\hat{\omega}_\ell}{2}\nu_a^2}}{Z(\hat{\omega}_\ell)} \\
&= \text{Log} \sum_{k=0}^{\infty} \frac{\hat{c}^k}{k!} \int \{d\hat{\pi}\}_k \left[\frac{Z(i\lambda_\varepsilon + \{\hat{\omega}\}_k)}{\prod_{\ell=1}^{k} Z(\hat{\omega}_\ell)}\right]^n \\
&\simeq \text{Log} \sum_{k=0}^{\infty} \frac{\hat{c}^k}{k!} \int \{d\hat{\pi}\}_k \left(1 + n \log \frac{Z(i\lambda_\varepsilon + \{\hat{\omega}\}_k)}{\prod_{\ell=1}^{k} Z(\hat{\omega}_\ell)}\right) \\
&= \text{Log}\, e^{\hat{c}} \left[1 + n \sum_{k=0}^{\infty} \frac{\hat{c}^k}{k!} e^{-\hat{c}} \int \{d\hat{\pi}\}_k \log \frac{Z(i\lambda_\varepsilon + \{\hat{\omega}\}_k)}{\prod_{\ell=1}^{k} Z(\hat{\omega}_\ell)}\right] \\
&\simeq \hat{c} + n \sum_{k=0}^{\infty} p_{\hat{c}}(k) \int \{d\hat{\pi}\}_k \log \frac{Z(i\lambda_\varepsilon + \{\hat{\omega}\}_k)}{\prod_{\ell=1}^{k} Z(\hat{\omega}_\ell)},
\end{aligned}
\tag{183}
$$

where $p_{\hat{c}}(k) = \frac{\hat{c}^k}{k!} e^{-\hat{c}}$ is a Poisson distribution with parameter $\hat{c}$. We remark that in the second line we have expressed $\exp(i\hat{\varphi}(\vec{\nu}))$ through its power series and a small $n$ expansion has been used across the entire calculation.

## F  The Kesten-McKay distribution from a peaked $\hat{\pi}$

We analytically derive the spectral density of the ensemble of adjacency matrices of random regular graphs (RRGs), using the formalism of Section 5. We employ Eq. (148) and (149), specialised to the RRG case where $p(k) = \delta_{k,c}$ and $p_K(K) = \delta(K-1)$. Therefore, we obtain for the self-consistency equation for $\hat{\pi}$

$$
\hat{\pi}(\hat{\omega}) = \int \{d\hat{\pi}\}_{c-1} \delta\left(\hat{\omega} - \frac{1}{i\lambda_\varepsilon + \sum_{\ell=1}^{c-1} \hat{\omega}_\ell}\right),
\tag{184}
$$

whereas for the spectral density we get

$$
\rho(\lambda) = \frac{1}{\pi} \lim_{\varepsilon \to 0^+} \text{Re} \int \{d\hat{\pi}\}_c \left[\frac{1}{i\lambda_\varepsilon + \sum_{\ell=1}^{c} \hat{\omega}_\ell}\right].
\tag{185}
$$

Eq. (184) can be solved by a degenerate pdf of the form

$$
\hat{\pi}(\hat{\omega}) = \delta(\hat{\omega} - \bar{\omega}_\varepsilon),
\tag{186}
$$

provided that $\bar{\omega}_\varepsilon$ solves

$$
\bar{\omega}_\varepsilon = \frac{1}{i\lambda_\varepsilon + (c-1)\bar{\omega}_\varepsilon} \Longleftrightarrow \bar{\omega}_\varepsilon = \frac{-i\lambda_\varepsilon \pm \sqrt{(i\lambda_\varepsilon)^2 + 4(c-1)}}{2(c-1)}.
\tag{187}
$$

Therefore, the spectral density reads

$$
\begin{aligned}
\rho(\lambda) &= \frac{1}{\pi} \lim_{\varepsilon \to 0^+} \mathrm{Re}\left[ \frac{1}{i\lambda_\varepsilon + c\bar{\omega}_\varepsilon} \right] \\
&= \frac{1}{2\pi} \lim_{\varepsilon \to 0^+} \mathrm{Re}\left[ \frac{(c-2)(i\lambda_\varepsilon) \mp c\sqrt{4(c-1)-\lambda_\varepsilon^2}}{\lambda_\varepsilon^2 - c^2} \right].
\end{aligned}
\tag{188}
$$

Taking the real part and thereafter the $\varepsilon \to 0^+$ limit in (188), one obtains

$$
\rho(\lambda) = \frac{c\sqrt{4(c-1)-\lambda^2}}{2\pi(c^2 - \lambda^2)} \quad \text{for } |\lambda| \le 2\sqrt{(c-1)},
\tag{189}
$$

where the minus sign has been chosen in order to have a physical solution. The latter expression is the Kesten-McKay pdf in Eq. (157).

# G   Trees have a symmetric spectrum

A tree is a connected acyclic undirected graph. Acyclic means that it contains no cycles. In a tree, any two nodes are connected via a unique path [87]. In particular, trees are examples of *bipartite* graphs, in which nodes can be divided into two disjoint subgraphs $S_1$ and $S_2$, such that every node in subgraph $S_1$ only has neighbours in the complementary subgraph $S_2$ and vice versa.

Here, we show that the $N \times N$ adjacency matrix $A$ (whether it is weighted or not) of a tree with $N$ nodes has a spectrum that is symmetric around $\lambda = 0$. In other words, for any eigenvalue $\lambda$ of $A$, then $-\lambda$ is also an eigenvalue of $A$. This result is encoded in the fact that in the set of recursion equations for the cavity inverse variances (29) and single-site inverse variances (31), the matrix entries appear only through their square.

Let $x$ be the eigenvector of $A$ corresponding to the eigenvalue $\lambda$. Given $x$ and $\lambda$, we will be able to construct a vector $y$ that is an eigenvector of $A$ corresponding to the eigenvalue $-\lambda$. Indeed, considering the eigenvalue equation for the component $x_i$,

$$
\lambda x_i = \sum_{j \in \partial i} A_{ij} x_j,
\tag{190}
$$

the node $i$ contributing to the l.h.s. of (190) and the nodes $\{j : j \in \partial i\}$ contributing to the r.h.s. of (190) always belong to different subgraphs. Therefore, the signs of all the components $x_j$ with $j \in \partial i$ all belonging to one of the two subgraphs ($S_1$ or $S_2$) can be changed, giving rise to

$$
-\lambda x_i = \sum_{j \in \partial i} (A_{ij})(-x_j) \Longleftrightarrow -\lambda y_i = \sum_{j \in \partial i} (A_{ij})(y_j).
\tag{191}
$$

This reasoning can be iterated for any $i = 1, \dots, N$. Therefore, given the eigenvector $x$ corresponding to $\lambda$, one can construct a vector $y$ such that

$$
y_i = \begin{cases} x_i & i \in S_1 \\ -x_i & i \in S_2 \end{cases}.
\tag{192}
$$

Of course, the choice of inverting the sign of the components of $x$ on the set $S_2$ is arbitrary. The same result is achieved by changing the sign of the components living on nodes in $S_1$ while leaving the components defined on nodes in $S_2$ unchanged. The vector $y$ is thus an eigenvector of the matrix $A$ corresponding to $-\lambda$.

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
