# Peer review of "Cavity and replica methods for the spectral density of sparse symmetric random matrices"

_SciPost Physics Lecture Notes, doi:SciPost Phys. Lect. Notes 33 (2021)_

## Round 2 · Referee Report · Anonymous (Referee 1) · 2021-3-5

Strengths

1) The paper clearly discusses the two main techniques in the field.

2) All calculations are very detailed.

3) There is a growing interest in the applications of random matrix theory to networks.

Weaknesses

1) The authors do not present a list of open problems or discuss promising research lines in the field.

2) Some sections present various technical details that are irrelevant for a person entering the field.

Report

In this paper the authors review the cavity method and the replica method to compute the spectral density of sparse (symmetric) random matrices. In my opinion, this work complies with its main purpose, i.e., to present a detailed account of the two main techniques to study the eigenvalue distribution of sparse random matrices. The present paper is a useful source of detailed information for students and young researchers that are entering the field of sparse random matrices. Thus, I recommend the present paper for publication in SciPost.

However, I have a couple of comments and suggestions that the authors should take into consideration before the paper is published.

1) I'd like to suggest a couple of references that would enrich the present manuscript. The cavity method for random matrices has been originally discussed in the following works:

P. Cizeau and J. P. Bouchaud, Phys. Rev. E 50, 1810 (1994)

A. Cavagna, I. Giardina and G. Parisi, Phys. Rev. Lett. 83 (1999)

Although the formulation of the cavity approach in the above papers is slightly different than in reference [31], the central ideas are already there. Concerning localization and the study of the IPR for undirected graphs, besides reference [41], the authors could also mention

G. Biroli, G. Semerjian and M. Tarzia, Prog. Theor. Phys. Suppl. 184, 187 (2010)

whose central ideas are similar to [41]. As far as I am aware, the first time that the replica-symmetric Gaussian ansatz appears in the context of sparse random matrices is the following work

D. S. Dean, J. Phys. A: Math. Gen. 35, L153 (2002)

Finally, the authors could be more precise when mentioning the difficulties with the Bray-Rodgers equation. I agree that a full analytic solution is not available, but this equation has been solved numerically in the above paper of Cavagna et al. In this context, there is also the recent work

https://arxiv.org/abs/2102.09629

which solves numerically a pair of equations that is analogous to eqs. (60) and (61) that yield the Bray-Rodgers equation.

2) Equation (2) has the merit to show that the spectral density follows from the averaged free energy of an analogous interacting system. However, the spectral density can be also computed from the imaginary part of the resolvent, and the cavity method can be directly applied to the resolvent matrix, which is a central object in random matrix theory. In my opinion, the cavity approach on the resolvent is more general, since the resolvent elements allow to compute other spectral observables and study, for instance, eigenvector localization. Given that the present paper is addressed to students and young researchers, I think it'd be interesting to briefly discuss the connection between the spectral density and the resolvent, to mention that the cavity method can be applied to the resolvent, and to discuss the connection between the complex variances of [31] and the diagonal part of the resolvent matrix.

3) I got a bit frustrated with the main outcome of section 4.4. The authors perform an involved perturbative expansion in powers of $1/c$, but in the end they present results only for the spectral density when $c \rightarrow \infty$. I think it'd be nice to show the $1/c$ correction to the Wigner law in eq. (117). If this is not possible, what is the point of expanding everything up to $O(1/c)$? I'd also suggest the authors make an effort to shorten section 4.4 (some equations, like eq. (105), can be written in a single line).

4) Since the paper is mostly addressed to those entering the field, it'd be useful to include in the final section a list of open problems or research lines that the authors find interesting to be further explored.

Minor comments:

1) One should mention that $c_{ii}=0$ around eq. (17).

2) The sentence "In this very sparse regime...thermodynamic limit is taken." on page 8 is a bit ambiguous. I suggest the authors reformulate it in a clearer way.

3) Please, specify that eq. (24) is exact only for $N \rightarrow \infty$.

4) Maybe it is a good idea to highlight in section 3.2 that the cavity equations have been rigorously proven in [32].

5) The authors introduce the spectral density of a single instance in eq. (32). I wonder whether this quantity has any interest, given that the spectral density of a finite system is a collection of $\delta$ peaks. In practice, one always ends up taking the limit $N \rightarrow \infty$. In addition, the rhs of eq. (32) is valid strictly for $N \rightarrow \infty$. Thus, in case the authors decide to keep eq. (32), it is better to highlight that this equation is an approximation for finite $N$.

6) It seems that a factor $N$ is missing in eq. (63).

7) I do not understand "which diverges for any finite $z$" just below eq. (100).

8) I found the explanation around eqs. (113) and (114) quite confusing. Maybe it is better to discuss how the spectral density transforms as soon as you introduce the change of variables, eqs. (87-89).

9) The authors write "These effects are much...degree $c$ increases" on page 31. It is not possible to draw this conclusion from figure 4, which presents results only for $\epsilon=10^{-3}$. It'd be interesting to complete figure 4 with results for $\epsilon=10^{-300}$. It'd be even better to see a graph of the point where the continuous spectrum vanishes as a function of $c$.

10) Section 6 discusses too many implementation details that, in my opinion, are not very helpful. The introduction of so many quantities to describe precisely how to implement the algorithm (in particular, the discussion about measurement sweeps) distracts the reader. Are $\mathcal{M}$ and $N_P$ the same quantity?

---

## Round 2 · Referee Report · Anonymous (Referee 2) · 2021-3-13

Strengths

1- very pedagogical and clearly written notes.

Weaknesses

1- lack of a broad overview on the motivations to study the problem and its applications. 2- lack of a perspective on the open problems and interesting new directions.

Report

The authors have written a pedagogical introduction to the replica and cavity methods for computing the spectrum of sparse random matrices.
The notes start from the Edward-Jones formula, and build progressively these tools. They focus on the Bray-Rodgers computation (section 4) and on the successive developments (section 5).
I found the notes very pedagogical and clearly written. I have only two main suggestions.
The problem, as well as the tools explained in the notes, have found applications in many different contexts. Despite this fact the motivation section (essentially the first six lines of page 3) is very short and not very useful for a young researcher starting to look at this problem. Correspondingly the bibliography appears to be rather short. I believe that it could be very useful to add a bit more of motivations/applications/bibliography.
The second suggestion concerns the conclusions. I think that it would be useful to mention the relevant open problems and to give some perspectives on the field.
Apart from this, I think that the notes meet the acceptance criteria and therefore I recommend them for publication in the “SciPost lecture notes” series.

---

## Round 3 · Referee Report · Anonymous (Referee 1) · 2021-5-25

Report

The authors have implemented several changes and the quality of the manuscript has improved in a significant way. In particular, the authors have added an extensive discussion of open problems in the final section, which is important to stimulate those entering in the field. I also appreciate very much the improvements in section 4.4, the inclusion of eq. (128), and the corresponding comparison with numerical diagonalization (figure 4). These explicit results are interesting and they may serve as an important guide for future studies of the spectral properties in the high connectivity limit.

I just have two remarks:

1) After eq. (128), the authors say that the normalization of the spectral density is fulfilled at any order in 1/c, since the 1/c correction is even. It gives the impression that this is a sufficient condition, which is not really precise. It is more correct to say that normalization implies that the integral of $\rho_1(x) $ over [-2,2] is zero. An analogous counterexample appears in the calculation of the 1/N correction to the spectral density of regular random graphs: the 1/N correction to the Kesten-McKay distribution is not an even function around x=0, but its integral is zero.

2) Below eq. (164), the authors argue that this equation may cease to be valid above the percolation transition, because of the presence of cycles. If this is correct and the tree-like assumption fails, why does the spectral density is well captured by the cavity equations above percolation? Besides that, there is plenty of evidence in other fields (spin-glasses, information science, etc) that the tree-like assumption holds above the percolation threshold, so the argument of the authors is not really supported by evidence.

  • validity: good
  • significance: good
  • originality: ok
  • clarity: good
  • formatting: good
  • grammar: good

Author:  Pierpaolo Vivo  on 2021-07-08  [id 1555]

(in reply to Report 2 on 2021-05-25)
Category:
reply to objection

Thank you for providing additional comments that we would like to address here and in the resubmitted version.

After eq. (128), the authors say that the normalization of the spectral density is fulfilled at any order in $1/c$, since the $1/c$ correction is even. It gives the impression that this is a sufficient condition, which is not really precise. It is more correct to say that normalization implies that the integral of $\rho_1(x)$ over [-2,2] is zero. An analogous counterexample appears in the calculation of the $1/N$ correction to the spectral density of regular random graphs: the $1/N$ correction to the Kesten-McKay distribution is not an even function around $x=0$, but its integral is zero.

We agree with the referee. To this purpose we have modified the main text after Eq. (128).

Below eq. (164), the authors argue that this equation may cease to be valid above the percolation transition, because of the presence of cycles. If this is correct and the tree-like assumption fails, why does the spectral density is well captured by the cavity equations above percolation? Besides that, there is plenty of evidence in other fields (spin-glasses, information science, etc) that the tree-like assumption holds above the percolation threshold, so the argument of the authors is not really supported by evidence.

In the paragraph below Eq. (164), we did not wish to imply that the cavity method fails entirely above the percolation threshold due to the presence of loops. Indeed its continued usefulness is clearly demonstrated by its ability to capture the spectral density of sparse matrices, as extensively demonstrated and discussed in the review. Instead, what we meant to be saying is that Eq. (164), i.e.

$$ {\rm Im}\, G(\lambda_\varepsilon){ii} = \mathrm{Re}\left[ \frac{1}{\omega_i}\right] = \pi \sum\alpha \delta_\varepsilon(\lambda-\lambda_\alpha) u_{i\alpha}^2 $$
may in the $\varepsilon\to0$-limit not allow one to correctly infer the statistics of squared eigenvector components $ u_{i\alpha}^2$, when evaluated using the cavity method in situations where it is only approximate, despite the fact that its summed version
$$ \rho_J(\lambda)=\lim_{\varepsilon\to 0^+}\frac{1}{\pi N} \mathrm{Im} \sum_{i=1}^N G(\lambda_\varepsilon)_{ii} \label{eq:sum_identity} $$
apparently does reproduce the spectral density rather well.

This need not be a contradiction, as the sum (2) may very well be robust against the approximations induced by the cavity method (which fails to be exact above the percolation threshold), even though its individual contributions Eq. (1) may not.

One possible, and indeed plausible mechanism is that an approximate evaluation of Eq. (1) using the cavity method may --- for quasi-degenerate eigenvalues in the continuous part of the spectrum --- effectively replace the eigenvector components appearing in Eq. (1) (and thus (2)) by components of vectors which are in fact superpositions or hybridizations of eigenvectors corresponding to several quasi-degenerate eigenvalues, i.e. by

$$ {\rm Im}\, G(\lambda_\varepsilon){ii} = \mathrm{Re}\left[ \frac{1}{\omega_i}\right] \simeq \pi \sum\alpha \delta_\varepsilon(\lambda-\lambda_\alpha) \tilde u_{i\alpha}^2\ , $$
in which the $\tilde u_{i\alpha}$ are linear superpositions of the form $\tilde u_{i\alpha} = \sum_{\mu \in I_\alpha} c_{\mu\alpha} u_{i\alpha}$. Assuming these superpositions to respect normalization, one immediately sees, that such a mechanism can induce even large discrepancies between the $u_{i\alpha}$ and the $\tilde u_{i\alpha}$ while leaving the spectral density as evaluated in terms of Eq. (2) unmodified due to orthonormality of the true eigenvectors.

We have modified the text after Eq. (164) to make this issue clearer in the revised manuscript.

---

## Round 3 · Referee Report · Anonymous (Referee 2) · 2021-6-8

Report

I recommed publication.

---

## Round 3 · Author Response

Dear Editor

we would like to resubmit our paper "Cavity and replica methods for the spectral density of sparse symmetric random matrices" for publication in Sci Post Lecture Notes series.

The referees provided interesting and valuable comments that helped us improve the quality of our work, therefore we wish to sincerely thank them.

We provide below a point-to-point reply to their reports, including a detailed list of changes made to the manuscript.

Referee 1

-->{References that would enrich the present manuscript. }

We thank Referee 1 for pointing out these references. We have incorporated them in the Introduction of the revised version of the paper and also in Section 4 when relevant. We believe that including these sources has greatly improved the breadth of our review.

--->{Connection between the spectral density and the resolvent.}

We followed the advice of the referee to include an explanation of the relation between the spectral density and the resolvent. To this purpose we have added Section 3.5 to the manuscript. We also establish the correspondence between the variables $\omega$ and the resolvent.
Moreover, in the Conclusions we suggest a way to study the square of eigenvector's components through the resolvent.

--->{Section 4.4: the $\mathcal{O}(1/c)$ correction to the Wigner semicircle. }

We thank the referee for raising this point. In fact, we noticed that the $\mathcal{O}(1/c)$ correction to the Wigner semicircle was already at hand in Section 4.4. Indeed, it was contained in the formula for the integral $I_A$ (now Eq. (112)).
Thus, we have amended Section 4.4 by including the explicit calculation of the $\mathcal{O}(1/c)$ term. We have also checked our analytical results against numerical diagonalisation. The section has been amended according to the other suggestions of the referee concerning this section. Moreover, we have added a final paragraph where we illustrate an alternative strategy for the evaluation of the large $c$ limit of the spectral density within the replica formalism. Although some further bits could be still shortened, we feel that keeping a more ``pedagogical" approach -- where more steps than strictly necessary are spelt out in full details -- may not hurt, especially the less experienced readership.

--->{Open problems in the final section.}

We have taken this comment fully on board. The second half of the Conclusions is now devoted to the descriptions of further research lines in the field. As mentioned before, we indicated a strategy for the investigation of eigenvectors through the resolvent formalism. Moreover, we also introduced the open problem of finite size effects at the localisation transition, providing relevant references. We also described another interesting pathway, represented by the study of the behaviour of the population dynamics algorithm at the transition.

Referee 2:

--->{Lack of broad overview on the motivations.}

We have taken the referee's comment fully on board. We have enriched the Introduction by illustrating various applications in physics of the spectral density of random matrix. In the same section we have also introduced the topic of Anderson localisation, and how the spectra of sparse random matrices have provided a framework for its study.
Consequently, and also thanks to Referee 1's suggestions, the bibliography has been expanded.

--->{Lack of a perspective on the open problems and interesting new directions.}

We have taken this comment fully on board. The second half of the Conclusions is now devoted to the descriptions of further research line in the field. As mentioned before, we indicated a strategy for the investigation of eigenvectors through the resolvent formalism. Moreover, we also introduced the open problem of finite size effects at the localisation transition, providing relevant references. We also described another interesting pathway, represented by the study of the behaviour of the population dynamics algorithm at the transition.

We trust that with these modifications the paper is now ready to be accepted for publication in Sci Post Lecture Notes series.

Vito Antonio Rocco Susca, Pierpaolo Vivo and Reimer K\"uhn

---

## Round 3 · List of Changes

--->{One should mention that $c_{ii}=0$ around eq. (17).

This has been done in the paragraph just before Eq. (17).

--->{The sentence "In this very sparse regime...thermodynamic limit is taken." on page 8 is a bit ambiguous. I suggest the authors
reformulate it in a clearer way.}

This has been done. The sentence has been replaced as follows.
"In this very sparse regime, the cavity method predictions are approximate for sparse graphs of finite size $N$, whereas they are exact for finite trees. However, the cavity results becomes asymptotically exact on finitely connected networks in the limit $N\to\infty$ (i.e. in
the thermodynamic limit)."

--->{Please, specify that eq. (24) is exact only for $N\to\infty$.}

This has been done right after Eq. (24).

--->{Maybe it is a good idea to highlight in section 3.2 that the cavity equations have been rigorously proven in [32].}

This has been done on page 8, right after the modified sentence quoted in point b).

--->{The authors introduce the spectral density of a single instance in eq. (32). I wonder whether this quantity
has any interest, given that the spectral density of a finite system is a collection of $\delta$-peaks. In
practice, one always ends up taking the limit $N\to\infty$. In addition, the rhs of eq. (32) is valid strictly for $N\to\infty$.
Thus, in case the authors decide to keep eq. (32), it is better to highlight that this equation is an approximation for finite $N$.}

Since in Section 3.2 we describe the single instance case, we have decided to keep Eq. (32). Following the referee's remark, we have added an explanatory note right after Eq. (32), though.

--->{ It seems that a factor $N$ is missing in eq. (63).}

The referee is right. We have corrected Eq. (63) (now Eq. (69)).

--->{ I do not understand "which diverges for any finite $z$" just below eq. (100).}

Right after Eq. (108) (former Eq. (100)), the quoted sentence has been replaced with "The series is divergent for any finite $z$".

--->{I found the explanation around eqs. (113) and (114) quite confusing. Maybe it is better to discuss
how the spectral density transforms as soon as you introduce the change of variables, eqs. (87-89).}

The discussion about the rescaling of the spectral density has been moved and more thoroughly discussed right after the change of variables. See Eq. (93) to (97) in the new version.

--->{The authors write "These effects are much...degree $c$ increases" on page 31. It is not possible
to draw this conclusion from figure 4, which presents results only for $\varepsilon=10^{-3}$. It'd be interesting to complete figure 4 with results for $\varepsilon=10^{-300}$. It'd be even better to see a graph of the point where the continuous spectrum vanishes as a function of $c$.}

This has been done. An extra plot can be found in Figure 4, where we compare the population dynamics results for $\varepsilon=10^{-3}$ with results for $\varepsilon=10^{-300}$ and results from numerical direct diagonalisation in logarithmic scale. The extra plot has been also commented on in the main text.

Moreover, in Figure 5 a plot comparing the continuous part of the spectrum for growing values of the mean degree $c$ has been added.

--->{ Section 6 discusses too many implementation details that, in my
opinion, are not very helpful. The introduction of so many
quantities to describe precisely how to implement the algorithm
(in particular, the discussion about measurement sweeps) distracts
the reader. Are $\mathcal{M}$ and $N_P$ the same quantity?}

We have clarified that $\mathcal{M}$ and $N_P$ are not the same quantity and are not related to each other. We have cleaned up part of the section on implementation details. We believe, however, that some of the
practical details provided in Section 6 may be useful as general instructions to help the reader build their own code. We believe the section is now more balanced.

---

## Round 4 · Author Response

Dear Editor,

we would like to resubmit our paper \textit{Cavity and replica methods for the spectral density of sparse symmetric random matrices} for publication in Sci Post Lecture Notes series.

After reviewing our revised version, one of the referees (Report 2 received 25th May 2021) has provided additional comments that we would like to address in this letter.

\begin{enumerate}

\item \textit{After eq. (128), the authors say that the normalization of the spectral density is fulfilled at any order in $1/c$, since the $1/c$ correction is even. It gives
the impression that this is a sufficient condition, which is not really precise. It is more correct to say that normalization implies that the integral of $\rho_1(x)$ over [-2,2] is zero. An analogous counterexample appears in the calculation of the $1/N$ correction to the spectral density of regular random graphs: the $1/N$ correction to the Kesten-McKay distribution is not an even function around $x=0$, but its integral is zero.}

We agree with the referee. To this purpose we have modified the main text after Eq. (128).

\item \textit{Below eq. (164), the authors argue that this equation may cease to be valid above the percolation transition, because of the presence of cycles. If this is correct and the tree-like assumption fails, why does the spectral density is well captured by the cavity equations above percolation? Besides that, there is plenty of evidence in other fields (spin-glasses, information science, etc) that the tree-like assumption holds above the percolation threshold, so the argument of the authors is not really supported by evidence. }

In the paragraph below Eq.\,(164), we did not wish to imply that the cavity method fails entirely above the percolation threshold due to the presence of loops. Indeed its continued usefulness is clearly demonstrated by its ability to capture the spectral density of sparse matrices, as extensively demonstrated and discussed in the review. Instead, what we meant to be saying is that Eq.\,(164), i.e.
\begin{equation}
{\rm Im}\, G(\lambda_\varepsilon)_{ii} = \mathrm{Re}\left[ \frac{1}{\omega_i}\right] = \pi \sum_\alpha \delta_\varepsilon(\lambda-\lambda_\alpha) u_{i\alpha}^2
\end{equation}
may in the $\varepsilon\to0$-limit not allow one to correctly infer the statistics of squared eigenvector components $ u_{i\alpha}^2$, when evaluated using the cavity method in situations where it is only approximate, despite the fact that its summed version
\begin{equation}
\rho_J(\lambda)=\lim_{\varepsilon\to 0^+}\frac{1}{\pi N} \mathrm{Im} \sum_{i=1}^N G(\lambda_\varepsilon)_{ii}
\label{eq:sum_identity}
\end{equation}
apparently does reproduce the spectral density rather well.

This need not be a contradiction, as the sum (2) may very well be robust against the approximations induced by the cavity method (which fails to be exact above the percolation threshold), even though its individual contributions Eq. (1) may not.

One possible, and indeed plausible mechanism is that an approximate evaluation of Eq.\,(1) using the cavity method may --- for quasi-degenerate eigenvalues in the continuous part of the spectrum --- effectively replace the eigenvector components appearing in Eq.\,(1) (and thus (2)) by components of vectors which are in fact superpositions or hybridizations of eigenvectors corresponding to several quasi-degenerate eigenvalues, i.e. by
\begin{equation}
{\rm Im}\, G(\lambda_\varepsilon)_{ii} = \mathrm{Re}\left[ \frac{1}{\omega_i}\right] \simeq \pi \sum_\alpha \delta_\varepsilon(\lambda-\lambda_\alpha) \tilde u_{i\alpha}^2\ ,
\end{equation}
in which the $\tilde u_{i\alpha}$ are linear superpositions of the form $\tilde u_{i\alpha} = \sum_{\mu \in I_\alpha} c_{\mu\alpha} u_{i\alpha}$. Assuming these superpositions to respect normalization, one immediately sees, that such a mechanism can induce even large discrepancies between the $u_{i\alpha}$ and the $\tilde u_{i\alpha}$ while leaving the spectral density as evaluated in terms of Eq. (2) unmodified due to orthonormality of the true eigenvectors.

We have modified the text after Eq.\,(164) to make this issue clearer in the revised manuscript.
\end{enumerate}

We trust that with these clarifications the paper is now ready to be accepted for publication in Sci Post Lecture Notes series.

Yours faithfully,

Vito Antonio Rocco Susca, Pierpaolo Vivo and Reimer K\"uhn

---

## Round 4 · List of Changes

see Authors comments

---

## Editorial Decision

published